# RcsF-independent mechanisms of signaling within the Rcs phosphorelay

**Anushya Petchiappan**[ID], **Nadim Majdalani**[ID], **Erin Wall**[ID][¤], **Susan Gottesman**[ID]*

Laboratory of Molecular Biology, Center for Cancer Research, National Cancer Institute, Bethesda, Maryland, United States of America

¤ Current address: US Food and Drug Administration, Office of Pharmaceutical Quality, Silver Spring Maryland, United States of America
* gottesms@mail.nih.gov

## Abstract

The Rcs (regulator of capsule synthesis) phosphorelay is a conserved cell envelope stress response mechanism in enterobacteria. It responds to perturbations at the cell surface and the peptidoglycan layer from a variety of sources, including antimicrobial peptides, beta-lactams, and changes in osmolarity. RcsF, an outer membrane lipoprotein, is the sensor for this pathway and activates the phosphorelay by interacting with an inner membrane protein IgaA. IgaA is essential; it negatively regulates the signaling by interacting with the phosphotransferase RcsD. We previously showed that RcsF-dependent signaling does not require the periplasmic domain of the histidine kinase RcsC and identified a dominant negative mutant of RcsD that can block signaling via increased interactions with IgaA. However, how the inducing signals are sensed and how signal is transduced to activate the transcription of the Rcs regulon remains unclear. In this study, we investigated how the Rcs cascade functions without its only known sensor, RcsF, and characterized the underlying mechanisms for three distinct RcsF-independent inducers. Previous reports showed that Rcs activity can be induced in the absence of RcsF by a loss of function mutation in the periplasmic oxidoreductase DsbA or by overexpression of the DnaK cochaperone DjlA. We identified an inner membrane protein, DrpB, as a multicopy RcsF-independent Rcs activator in *E. coli*. The loss of the periplasmic oxidoreductase DsbA and the overexpression of the DnaK cochaperone DjlA each trigger the Rcs cascade in the absence of RcsF by weakening IgaA-RcsD interactions in different ways. In contrast, the cell-division associated protein DrpB uniquely requires the RcsC periplasmic domain for activation; this domain is not needed for RcsF-dependent signaling. This suggests the possibility that the RcsC periplasmic domain acts as a sensor for some Rcs signals. Overall, the results add new understanding to how this complex phosphorelay can be activated by diverse mechanisms.

## Author summary

The Rcs phosphorelay signaling cascade regulates the expression of genes related to capsule synthesis, biofilm formation, virulence, and cell division in enterobacteria and is

**Data Availability Statement:** All data is either in main paper or supplemental material. SI Data files contain prism files with all of the underlying data for graphs in the paper.

**Funding:** Funding was provided by Intramural funding from the Center for Cancer Research, NCI, NIH to SG, AP, NM and EAW; EAW was supported by an NIGMS Prat Fellowship. The funders had no role in study design, data collection and analysis, decision to publish, or preparation of the manuscript.

**Competing interests:** The authors have declared that no competing interests exist.

critical for cell membrane integrity and response to beta-lactam antibiotics and antimicrobial peptides. RcsF is the sole known sensor, but other proteins have been reported to activate this pathway in the absence of RcsF. We have discovered a novel RcsF-independent Rcs activator and found that each of three RcsF-independent proteins activate the system differently. Most significantly, we find that the histidine kinase RcsC can be involved in signal sensing independently of RcsF. Our study sheds light into the complex mechanisms of Rcs activation and adds to our knowledge of non-orthodox signaling systems across organisms.

## Introduction

The Gram-negative bacterial cell wall envelope comprises an outer membrane, the periplasm, a peptidoglycan layer, and an inner membrane [1]. It serves as a protective barrier against environmental insults and a permeability barrier for selective uptake of nutrients. Due to its structural and functional importance for growth and metabolism, bacteria have evolved a multitude of envelope stress response systems to monitor various stresses at the cell wall and respond to them in a timely manner. These systems typically contain a sensor protein which detects the stressor and subsequently activates the downstream signaling pathway, leading to changes in gene transcription. A highly conserved envelope stress response pathway in enterobacteria is the Rcs phosphorelay (reviewed in [2,3]). This cascade responds to outer membrane and peptidoglycan stress from both external and intrinsic sources. While Rcs is a member of the ubiquitous histidine kinase/response regulator signaling systems, in which phosphorylation of the response regulator regulates output, it is significantly more complex than the canonical two-component signaling systems, making it a unique but challenging model to study non-orthodox signaling systems across organisms.

The Rcs pathway has been shown to be activated in response to beta-lactams, antimicrobial peptides, osmotic shock, acid stress, defects in LPS trafficking, peptidoglycan biosynthesis, and contact with a solid surface (reviewed in [2,3]). The Rcs regulon was first defined for its role in regulating synthesis of capsular polysaccharide but has been shown to include genes related to biofilm formation, motility, virulence, cell morphology, and cell division, among others (reviewed in [2]). How Rcs responds to a wide array of inducers, and the exact mechanism of signal sensing and transduction remains unclear; structural information about most of the components is currently limited or lacking [4–9].

The multi-step Rcs phosphorelay comprises an inner membrane hybrid histidine kinase RcsC, an inner membrane phosphotransfer protein RcsD, and the response regulator RcsB (Fig 1A). The sensor for this cascade is an outer membrane lipoprotein RcsF [10–12]. RcsF detects LPS/peptidoglycan defects and chemical stressors like polymyxin B (presumed to disrupt LPS), A22 (a MreB inhibitor), and mecillinam (a beta-lactam antibiotic) [13,14]. The phosphorelay is negatively regulated by inner membrane protein IgaA. IgaA is essential and it can only be deleted if the phosphorelay is inactivated by mutations in *rscC*, *rcsD*, or *rcsB* [15]. During normal growth, IgaA represses signaling by interacting with RcsD. When RcsF perceives stress at the envelope, it activates signaling by interacting with the periplasmic domain of IgaA [16,17]. This relieves the repression of signaling by IgaA, presumably changing the way in which RcsD interacts with RcsC to induce the phosphorelay [18]. A required step in the phosphorelay activation is the autophosphorylation of RcsC. RcsC is a complex histidine kinase, with a receiver domain with a conserved aspartate at its C-terminus (Fig 1A). A phosphate group is transferred from the conserved histidine residue in RcsC to the aspartate within

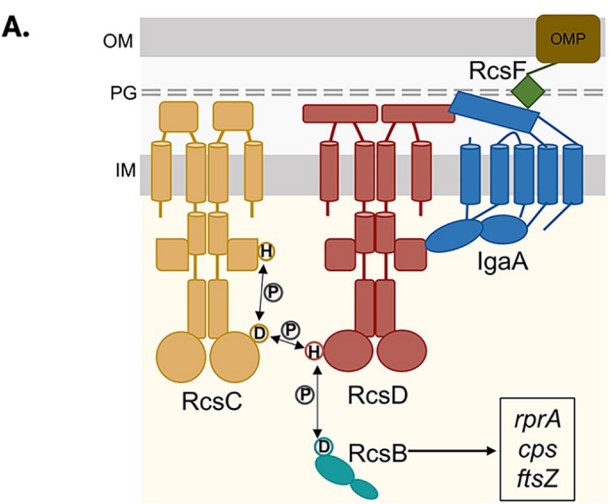

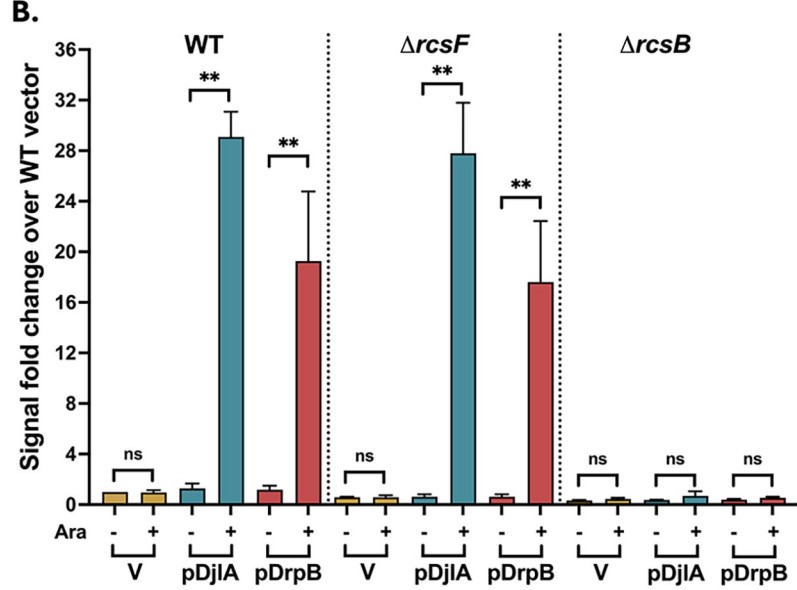

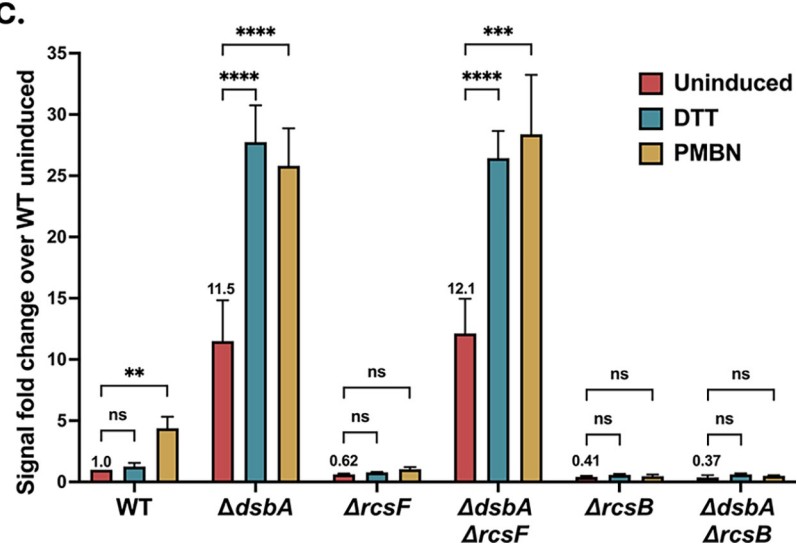

**Fig 1. RcsF-independent activators of Rcs signaling. A. Components of the Rcs phosphorelay:** The Rcs pathway is comprised of the hybrid histidine kinase RcsC, the phosphotransferase RcsD, and the response regulator RcsB as well as the upstream regulatory components RcsF and IgaA. Under non-inducing growth conditions, IgaA represses signaling by interacting with RcsD. RcsF senses damage to the envelope and triggers the RcsC-RcsD-RcsB phosphorelay by interacting with IgaA, altering its interaction with RcsD. **B. Rcs activation by overexpression of DjlA and DrpB:** All strains carry a *rprA* promoter fusion to mCherry ($P_{rprA}$::mCherry); mCherry fluorescence acts as an indicator for Rcs activation. For the $P_{rprA}$::mCherry assay, the strains overexpressing DjlA (pBAD-DjlA/pPSG961) or DrpB (pBAD-DrpB/pDSW1977) were grown in MOPS minimal glycerol medium containing chloramphenicol (25 μg/ml) and either 0.2% glucose or 0.02% arabinose at 37°C. The RFU at OD 0.4 compared to the uninduced vector control (set to 1) is plotted. The strains used are: WT (EAW8), *rcsF*::*kan* (AP51), and *rcsB*::*kan* (EAW31). **C. Rcs signaling in *dsbA* mutants:** For the $P_{rprA}$::mCherry assay, the cells were grown in MOPS minimal glucose medium at 37°C. The RFU at OD 0.4 as compared to WT uninduced, set to 1, is depicted here. The cells were treated with either 1mM DTT (blue bars) or 20μg/ml PMBN (brown bars) from the beginning of growth. The strains used were: WT (EAW8), *rcsF*::*cat* (EAW32), *rcsB*::*kan* (EAW31), *dsbA*::*kan* (EAW62), *dsbA*::*kan rcsF*::*cat* (EAW67), and Δ*dsbA rcsB*::*kan* (AP12). Details of the assay are described in Materials and Methods. Data from three independent experiments are plotted as mean with error bars indicating the standard deviation. Values were statistically analyzed using multiple unpaired *t*-tests. Statistical significance is indicated as follows: ns ($P > 0.05$; non- significant), * ($P < 0.05$), ** ($P \leq 0.01$), *** ($P \leq 0.001$), and **** ($P \leq 0.0001$).

the RcsC receiver domain. This phosphate group is then transferred to a conserved histidine residue in RcsD and subsequently to an aspartate in RcsB [19,20]. Phosphorylated RcsB dimers bind promoters to regulate transcription. Alternatively, RcsB can act in concert with other auxiliary partners like RcsA to regulate gene expression (reviewed in [2]). IgaA and RcsD contact each other both in the periplasm and in the cytoplasm; weakening of either interaction leads to activation of the phosphorelay [18]. Thus, IgaA provides a braking mechanism, with the cytoplasmic contact with RcsD as the regulatory switch and the periplasmic interaction preventing Rcs over-activation. The identification of the cytoplasmic interaction as the likely regulatory switch was further underscored by our isolation of a dominant mutation in *rcsD* in its cytoplasmic PAS-like domain (T411A) [18]. The RcsD T411A mutation blocks induction by polymyxin B nonapeptide (PMBN) and other inducers due to tightened cytoplasmic interaction with IgaA [18].

While the histidine kinase is the sensor for most two-component systems, RcsC is not known to be a sensor for any of the known Rcs signals. Consistent with this, we previously demonstrated that the RcsC periplasmic domain is dispensable for polymyxin B nonapeptide (PMBN) induction of the Rcs cascade [18]. The outer membrane lipoprotein RcsF remains the only known sensor of the system. RcsF resides within the lumen of the OMP such that a portion of it is surface-exposed [12,21,22]. RcsF activates Rcs upon sensing signals perturbing the cell surface and the peptidoglycan layer or upon its mislocalization to the inner membrane (reviewed in [2]).

While the majority of inducing signals require RcsF, there are a number of situations in which RcsF-independent activation of Rcs has been reported. Overexpression of the DnaK cochaperone protein DjlA activates Rcs even in the absence of RcsF [23,24]. Loss-of-function mutations in *dsbA*, encoding a periplasmic oxidoreductase, also activate the phosphorelay independently of RcsF [11]. Overproduction of the response regulator RcsB is sufficient to activate downstream targets [25]; work in *Salmonella* identified the phosphorelay protein BarA as an RcsF-independent activator of RcsB [26]. Overproduction or mutation of a number of proteins, including TolB [27], DrpB [28], YpdI [29], YmgABC [30], YfgM [31,32] and YqjA [33], have been linked with Rcs activation but the role of RcsF for activation in these cases has not been reported [11]. Given the complexity of the Rcs phosphorelay, it would not be surprising if there are alternative methods of signaling for Rcs activation; RcsF-independent activation may reflect the existence of such additional signaling pathways.

Here, we further investigated the nature of RcsF-independent activation in *dsbA* mutants and upon *djlA* overproduction. We performed a screen for novel RcsF-independent Rcs

activators and uncovered a third pathway for RcsF-independent signaling. Comparisons between these pathways delineated three genetically distinct modes of Rcs induction, demonstrating that they are also mechanistically distinct, revealing novel aspects of the regulation of the phosphorelay and implicating the RcsC periplasmic domain in one of these signaling pathways.

## Results

### DrpB is a novel RcsF-independent Rcs activator

With the aim of identifying novel proteins capable of triggering the Rcs cascade in an RcsF-independent manner, we utilized a fluorescence-based genetic screen in *E. coli*. We transformed a plasmid library carrying 3–5 kb fragments of the *E. coli* chromosome into strain EAW34, carrying a reporter for Rcs activation and deleted for *rcsF*. The reporter used is a transcriptional fusion of the *rprA* promoter, an RcsB target, to the mCherry protein, referred to here as $P_{rprA}$::mCherry [18,34]. This strain normally shows low fluorescence in the absence of RcsF, consistent with a central role for RcsF in Rcs activation. The colonies were screened for increased fluorescence and 14 plasmids containing putative RcsF-independent Rcs activators were isolated. Sequencing of the plasmids revealed that 13 of them contained overlapping fragments of the *E. coli* chromosome that included genes encoding the YedR (DrpB) ORF along with the RseX sRNA (S1A Fig). The remaining plasmid contained genes encoding the YihA protein and the sRNAs Spot 42 and CsrC (S1B Fig). We previously reported that overexpression of Spot42 induces the Rcs pathway and while that study suggested that much of the Spot 42 effect was RcsF dependent, only a qualitative assay was done [34]. Spot 42, an Hfq-dependent regulatory RNA, regulates translation and stability of multiple mRNAs and has pleiotropic effects in the cell [35,36], making identification of a single critical target complicated.

To further validate identification of DrpB as capable of RcsF-independent induction, we assayed the growth and fluorescence of wildtype, Δ*rcsF* and Δ*rcsB* strains expressing DrpB or the known RcsF-independent activator DjlA under control of the arabinose-inducible pBAD promoter, using the same $P_{rprA}$::mCherry fusion as a reporter for Rcs activation. Overexpression of both DrpB and DjlA showed increased fluorescence, both in the presence and absence of RcsF, confirming their role as multicopy RcsF-independent Rcs activators (Fig 1B). The increased fluorescence was fully dependent on RcsB for both genes. We did not observe a significant RcsF-independent increase in the reporter expression after overexpression of YihA or *rseX*, and the modest induction by Spot42 in Δ*rcsF* was also not significant (S1C and S1D Fig). Therefore, we focused our investigation on DrpB. DrpB is a small inner membrane protein and was previously found to activate both the Rcs and the Psp stress response upon overexpression, suggesting that its overexpression leads to membrane stress [28]. We first compare DrpB to other RcsF-independent activators before examining it in more detail.

We also confirmed that the deletion of *dsbA* activates the Rcs pathway in an RcsF-independent manner, by measuring the fluorescence of the same $P_{rprA}$::mCherry reporter in Δ*dsbA*, Δ*dsbA* Δ*rcsF*, and Δ*dsbA* Δ*rcsB* strains (Fig 1C). Deletion of *dsbA* increased the basal $P_{rprA}$::mCherry activity independently of RcsF but not RcsB. Strains devoid of DsbA are defective in disulfide bond formation in periplasmic proteins and treatment with the reducing agent DTT, by further interfering with disulfide bond formation, exacerbates this problem [37]. Addition of DTT further induced the Rcs pathway, but only in the absence of DsbA. Unexpectedly, we found that PMBN, which normally induces the pathway in an RcsF-dependent-fashion (see WT and Δ*rcsF* results in Fig 1C), also induced this pathway, to a similar extent as DTT, in both Δ*dsbA* and Δ*dsbA* Δ*rcsF* strains (Fig 1C). Together, these results provide evidence for three

RcsF-independent Rcs activation situations–lack of DsbA and overproduction of DjlA or DrpB.

We next investigated whether these inducers of Rcs depend on each other. We first measured Rcs activation in *dsbA* mutants in cells also deleted for the chromosomal copies of *djlA* or *drpB* (S2A Fig). Activity in the Δ*dsbA* Δ*djlA* and Δ*dsbA* Δ*drpB* strains was essentially unchanged from that seen in the Δ*dsbA* strain (see Fig 1C). Therefore, the absence of DsbA does not act by mimicking overexpression of DjlA or DrpB. In addition, overexpression of DjlA or DrpB induces activity in a *dsbA* deletion strain (S2B Fig), although the overexpression of DrpB led to poor growth in the absence of DsbA (see OD values, S2B Fig). We also tested if the multicopy activators DjlA and DrpB require each other to trigger Rcs signaling and found that they did not (S2C Fig). Together, these results indicate that DsbA, DjlA, and DrpB do not require one another for activity and possibly employ different mechanisms of Rcs activation. We next investigated the pathways they each use for RcsF-independent activation of Rcs.

## DrpB has a unique requirement for the RcsC periplasmic domain for activation

The Rcs pathway can be activated simply by increasing the levels of RcsB [25]. If this were how the RcsF-independent factors were working, they might be independent of the histidine kinase, RcsC and the phosphotransfer protein RcsD, as observed for the effect of mutations in *barA* [26]. Previous work had demonstrated that multicopy DjlA and DrpB are still dependent on RcsC for signaling [23,24,28]. As seen in Fig 2, while deletion of *rcsC* caused an increase in the basal level of P_{rprA}::mCherry expression, there was no further induction in the absence of RcsC for any of these factors. Deletion of *rcsC* or of *rcsD* leads to higher basal levels of expression of the reporter, interpreted as reflecting loss of phosphatase activity combined with some phosphorylation of RcsB from other sources [18,34].

We previously showed that the RcsC periplasmic domain is not needed for the RcsF-dependent induction by PMBN [18]. We assayed Rcs activation upon deletion of *dsbA* or overexpression of DjlA or DrpB in strains encoding a chromosomal version of RcsC missing its

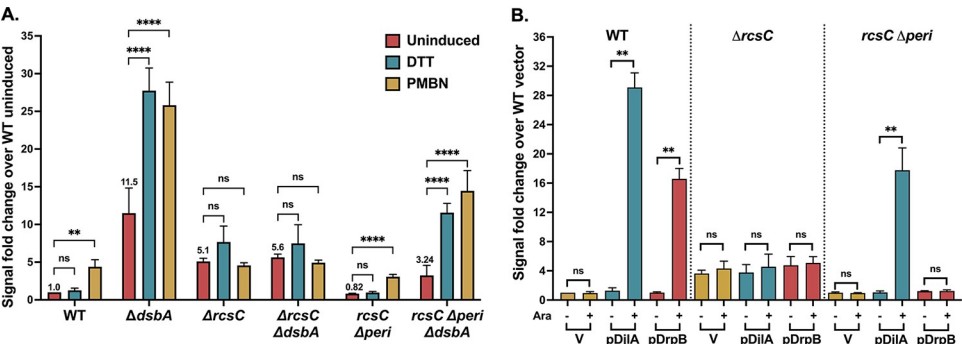

**Fig 2. Requirement of RcsC periplasmic domain for RcsF-independent signaling by DrpB. A. *dsbA* mutants do not need the RcsC periplasmic domain for signaling:** For the P_{rprA}::mCherry assay, the cells were grown in MOPS minimal glucose medium at 37°C. The RFU at OD 0.4 as compared to WT uninduced, set to 1, is depicted here. The cells were treated with either 1mM DTT or 20µg/ml PMBN. **B. DrpB, but not DjlA, requires the RcsC periplasmic domain for Rcs activation:** For the P_{rprA}::mCherry assay, the strains overexpressing DjlA (pBAD-DjlA/pPSG961) or DrpB (pBAD-DrpB/pDSW1977) were grown in MOPS minimal glycerol medium containing chloramphenicol (25 µg/ml) and either 0.2% glucose (-Ara) or 0.02% arabinose at 37°C. The RFU at OD 0.4 compared to the uninduced vector control is plotted. The strains used were: WT (EAW8), *dsbA*::*kan* (EAW62), *rcsC*::*tet* (EAW18), *rcsC*::*tet dsbA*::*kan* (EAW63), *rcsC*Δ*peri* (EAW70), and *dsbA*::*kan rcsC*Δ*peri* (EAW74). Values (mean ± SD) were statistically analyzed using multiple unpaired *t*-tests. Statistical significance is shown as: ns (P > 0.05; non- significant), ** (P ≤ 0.01), and **** (P ≤ 0.0001).

periplasmic domain (*rcsCΔperi*). Fig 2A shows that PMBN was able to induce signaling in the *rcsC Δperi* strain as previously found. Deletion of *dsbA* in the *rcsCΔperi* strain increased the basal level of signaling, although not to the extent seen for the *rcsC*⁺ strain (Δ*dsbA*); DTT or PMBN further increased signaling (Fig 2A). These results demonstrate that the primary effect of Δ*dsbA* is independent of the RcsC periplasmic region. It seems likely that the loss of *dsbA* has additional effects in the periplasm, and this may contribute to the muted response in signal in the *rcsCΔperi* strain (Fig 2A). It is also likely that the RcsCΔperi protein is somewhat perturbed in activity, for instance somewhat reducing the balance of kinase to phosphatase, leading to less activity of the reporter. DjlA overexpression was also able to activate signaling in the *rcsCΔperi* strain, although at a somewhat decreased level (Fig 2B). However, upon DrpB overexpression, no Rcs activation was observed in the *rcsCΔperi* strain (Fig 2B). This is the first evidence of a role for the RcsC periplasmic domain in activation of the phosphorelay and provides clear evidence that DrpB acts in a manner that is distinct from DjlA overexpression or loss of DsbA.

## DjlA, but not DsbA and DrpB, can induce Rcs signaling in a RcsD T411A mutant

As for deletion of *rcsC* (Fig 2), all three factors were unable to induce Rcs activity in the absence of the phosphorelay protein RcsD (Fig 3). RcsD T411A is an allele of RcsD expressing a mutant in the cytoplasmic domain of RcsD that has a higher affinity for IgaA, blocking RcsF-dependent PMBN signaling [18] (Fig 3A). Strains carrying *rcsD* T411A were used to further explore the role of IgaA-RcsD interactions in RcsF-independent activation. The increase in the reporter in the absence of DsbA signaling was blocked by the *rcsD* T411A mutation; the basal level was reduced to that seen in *rcsD* T411A alone, and the DTT or PMBN increase was blocked (Fig 3A). Overexpression of DrpB also could not induce reporter activity in the *rcsD* T411A strain (Fig 3B). However, DjlA overexpression was still able to activate reporter

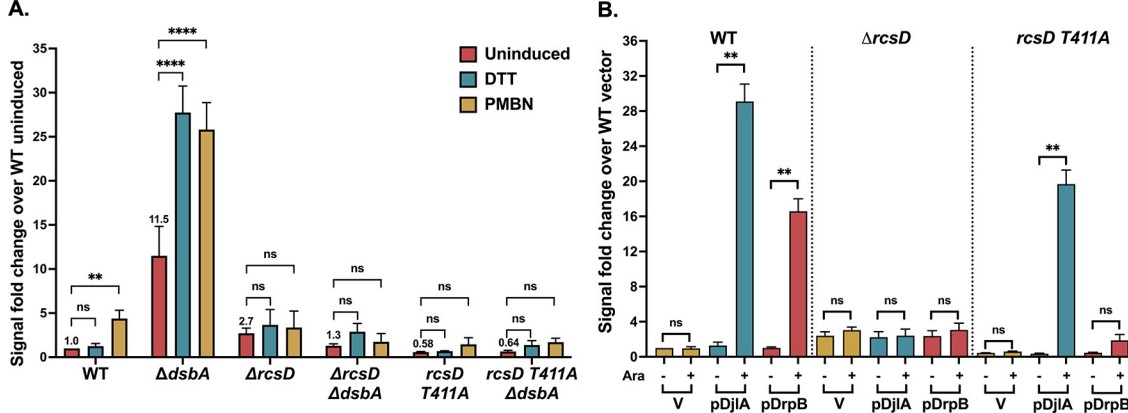

**Fig 3. Effect of RcsD T411A mutation on RcsF-independent signaling. A. RcsD T411A mutation blocks induction by *dsbA*:** For the $P_{rprA}$::mCherry assay, the cells were grown in MOPS minimal glucose medium at 37°C. The RFU at OD 0.4 as compared to the uninduced WT control, set to 1, is depicted here. The cells were treated with either 1mM DTT or 20μg/ml PMBN. Details of the assay are described in Materials and Methods. **B. DjlA, but not DrpB, can overcome the RcsD T411A mutation for Rcs activation:** For the $P_{rprA}$::mCherry assay, the strains overexpressing DjlA (pDjlA/pPSG961) or DrpB (pDrpB/pDSW1977) were grown in MOPS minimal glycerol medium containing chloramphenicol (25 μg/ml) and either 0.2% glucose or 0.02% arabinose at 37°C. The RFU at OD 0.4 compared to the vector uninduced control, set to 1, is plotted. The strains used were: WT (EAW8), *dsbA*::*kan* (EAW62), Δ*rcsD* (EAW19), Δ*dsbA rcsD541*::*kan* (AP13), *rcsD T411A* (EAW121), and *rcsD T411A dsbA*::*kan* (AP14). Statistical significance was calculated using multiple unpaired *t*-tests and is shown as follows: ns ($P > 0.05$; non-significant), * ($P < 0.05$), ** ($P \leq 0.01$), and **** ($P \leq 0.0001$).

expression in this strain, indicating its unique ability to overcome the effect of the *rcsD* T411A mutation.

We considered the possibility that DjlA could function in the *rcsD* T411A strain because it was the strongest inducer of the conditions tested. As seen in Fig 1B and 1C, DjlA overexpression gave around 30-fold induction, which is higher than DrpB overexpression (~17 fold), DsbA deletion (~13 fold), and the RcsF-dependent response to PMBN (~4 fold). As a test of this model, we examined the ability of RcsD T411A to block activity when RcsF is highly active. Overproduction of RcsF or expression of a derivative (RcsF$^{IM}$, carrying mutations S17D and M18Q) that localizes the protein to the inner member and constitutively activates Rcs signaling [14,38] both induce Rcs activity, with the inner-membrane localized mutant RcsF derivative, RcsF$^{IM}$, being the stronger activator, inducing better than 25-fold (S3 Fig). However, the T411A mutation was still able to block this induction (S3 Fig). These results suggest that DjlA is unique in its ability to fully overcome the *rcsD* T411A allele.

Thus, each of the three RcsF-independent factors has distinct patterns of induction, suggesting that they act in different manners. While all three are dependent upon RcsC and RcsD, only DrpB depends on the RcsC periplasmic domain and only DjlA can bypass the *rcsD* T411A allele to induce Rcs activity. Below we further examine how each of these factors works.

## DsbA is needed for efficient IgaA-RcsD periplasmic interactions

DsbA resides in the periplasm and plays a central role in disulfide bond (DSB) formation in *E. coli* and other bacteria [reviewed in [39]]. DsbA, in concert with DsbB, introduces disulfide bonds into newly synthesized periplasmic proteins, predominantly between consecutive cysteine residues [40–42]. *dsbA* null mutants show a pleotropic phenotype consistent with defects in the cell envelope—they are mucoid, non-motile, sensitive to DTT as well as some drugs, show decreased fitness and attenuated virulence [37,39,40,43]. Periplasmic and membrane proteins that require disulfide bonds frequently require DsbA for proper folding. One such protein is the Rcs sensor protein, the outer membrane lipoprotein RcsF [44]. RcsF has two non-consecutive disulfide bridges and impaired disulfide bond formation leads to its degradation [5,7]. Thus, RcsF is non-functional in a *dsbA* mutant, supporting the idea that mucoidy in the *dsbA* mutant strain, likely reflecting Rcs induction, cannot be due to activation via RcsF; this is confirmed by the induction of the P$_{rprA}$::mCherry reporter in the absence of RcsF in Fig 1C. This is further validated by the absence of Rcs induction in response to Mecillinam or the MreB inhibitor A22 by the *dsbA* mutants, both dependent on RcsF (S4A Fig). Note that this result also distinguishes PMBN from these other inducers as uniquely still able to induce the Rcs phosphorelay in the *dsbA* mutant (Figs 2A and S4B; compare differences in fold induction).

Aside from RcsF, three other proteins in the phosphorelay (RcsC, RcsD, and IgaA) have a periplasmic domain. However, RcsD does not have any conserved cysteine residues within its periplasmic domain, ruling it out as a direct target of DsbA.

RcsC has two conserved cysteine residues in its periplasmic domain, but since the Δ*dsbA* *rcsCΔperi* strain is still DTT-inducible (Fig 2A), this cannot be the critical target for DsbA. However, our results do suggest that, directly or indirectly, the RcsC periplasmic domain contributes to phosphorelay activation in the absence of DsbA. While deletion of the RcsC periplasmic domain had only a modest effect on P$_{rprA}$::mCherry basal level or induction in a *dsbA*$^{+}$ strain (compare WT to *rcsCΔperi* data, Fig 2A), levels of expression in the Δ*dsbA* mutants were significantly lower for Δ*dsbA* *rcsCΔperi* than for Δ*dsbA* *rcsC*$^{+}$ (Fig 2A), with Δ*dsbA* 12-fold higher than WT in the absence of induction, while Δ*dsbA* *rcsCΔperi* was only 3.2-fold increased compared to WT or *rcsCΔperi* (Fig 2A). To determine if this reduction in induction

was dependent upon RcsC disulfide bonds, we examined the behavior of a strain expressing RcsC mutant for the C111 and C154 periplasmic cysteine residues (*rcsC C111A C154A*, named here as *rcsC C2A*) in *dsbA⁺* or Δ*dsbA* strains. As seen in S4B Fig, the *rcsC C2A dsbA⁺* strain has the same pattern of expression as the WT strain or a *rcsCΔperi* strain (Fig 2A), responding to PMBN but not DTT. The Δ*dsbA rcsC C2A* strain, unlike the *rcsCΔperi* strain, responds to both DTT and PMBN and this response is not muted as in *rcsCΔperi* (compare to Fig 2A). This suggests the effect of the *rcsCΔperi* does not reflect a role of the RcsC periplasmic cysteine residues and is likely more indirect; this was not further explored here.

IgaA has four conserved periplasmic cysteine residues, suggesting that there are two disulfide linkages in its periplasmic domain. A study in *Salmonella* supported the idea that at least the first of the disulfide bonds was important for proper IgaA function [45]. Given that IgaA is essential, the *dsbA* mutant cannot lead to fully non-functional IgaA. However, if the disulfide bonds are not properly formed, the periplasmic interaction between IgaA and RcsD could be affected in the absence of DsbA, leading to the observed Rcs activation. We have previously shown that IgaA and RcsD interact in a bacterial two hybrid (BACTH) assay, and that the interaction detected by this assay is primarily dependent upon a periplasmic contact between the proteins [18]. This assay is based on the reconstitution of the *Bordetella pertussis* adenylate cyclase from its two fragments, T18 and T25, in an adenylate cyclase mutant (*cya*) *E. coli* strain BTH101 [46,47]. Fusion of each of these fragments to an interacting protein reconstitutes adenylate cyclase, assayed by the Cya-dependent synthesis of beta-galactosidase. Both orientations, IgaA-T18/RcsD-T25 as well as IgaA-T25/RcsD-T18, have been demonstrated to show a strong interaction ([18] and Fig 4A, WT strain).

We introduced the *dsbA* null mutant allele into the host for the BACTH assay and studied its effect on the IgaA-RcsD interaction, compared to the WT strain (Fig 4A). The interaction between IgaA and RcsD was significantly impaired in the *dsbA* mutant, suggesting that DsbA is needed to maintain a proper IgaA-RcsD interaction, possibly due to its role in IgaA folding. Given the localization of DsbA in the periplasm, we would expect that it is the periplasmic interaction of IgaA with RcsD that should be defective in the absence of DsbA. This was confirmed first by testing the interaction of RcsD with various domain deletion constructs of IgaA. IgaA has five transmembrane helices anchoring two cytoplasmic domains and a large periplasmic domain (schematic in Fig 4B). The IgaAΔ*peri* (Δ384–649) protein has been shown to have a weak interaction with RcsD, confirmed here; loss of DsbA did not further decrease the interaction (Fig 4B). Deletion of the two cytoplasmic domains (IgaAΔcyt), on the other hand, did not decrease the interaction, but was very sensitive to loss of DsbA (Fig 4B). Neither of these deletions significantly affected the levels of expression of the IgaA-T18 protein, in the presence of absence of DsbA (S4D Fig). This data is best explained by a change in the IgaA periplasmic domain in the absence of DsbA, reducing the periplasmic interaction with RcsD.

A crystal structure of the *E. coli* IgaA periplasmic domain shows two disulfide linkages, C404-C425 and C498-C504, consistent with previous work [4,45]. We mutated the Cysteine residues to Serine and tested their effect on the IgaA-RcsD interaction, either in pairs (C404S C425S or C498S C504S) or in a mutant with all four mutated (C4S) (S4C Fig). Breaking either linkage significantly weakened the interaction of IgaA with RcsD. As with the data for Δ*dsbA* in Fig 4B, the RcsD T411A allele overcame much (but not all) of the loss of interaction in the IgaAC4S derivative (S4C Fig). Therefore, for this interaction assay, C4S mimics the effect of loss of *dsbA*, supporting the idea that the disulfide linkages are required for proper folding of the periplasmic domain of IgaA and its periplasmic interaction with RcsD. Mutations in IgaA may lead to its degradation during the stationary phase [45] but the overexpressed IgaA C4S mutant was stable in the T18 fusion, suggesting DsbA primarily affects its folding (S4D Fig).

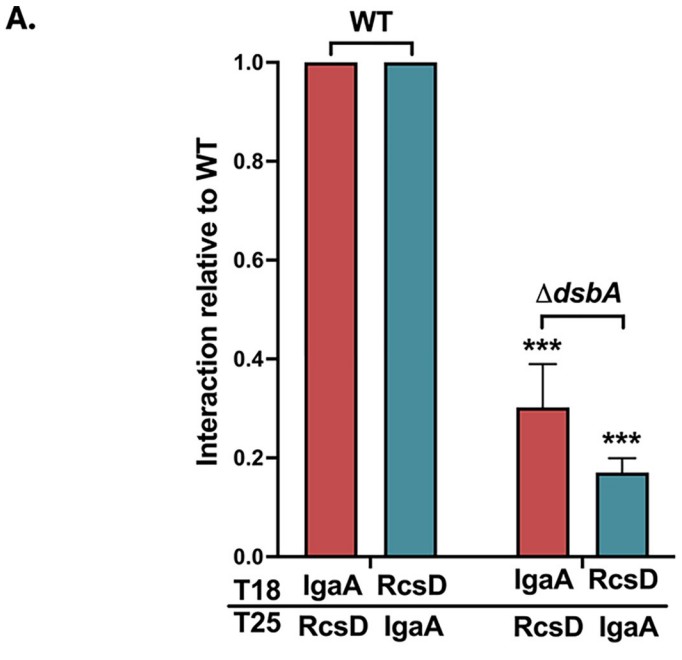

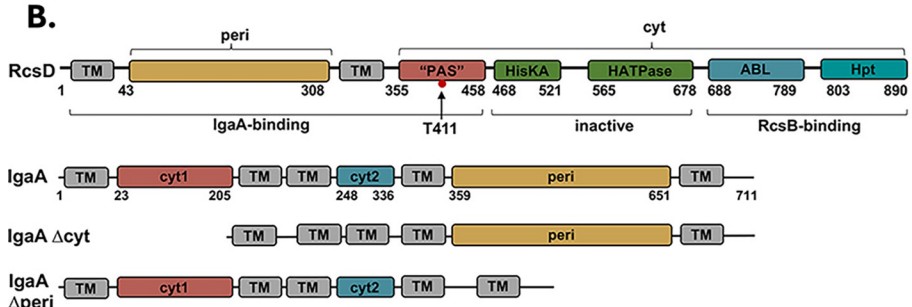

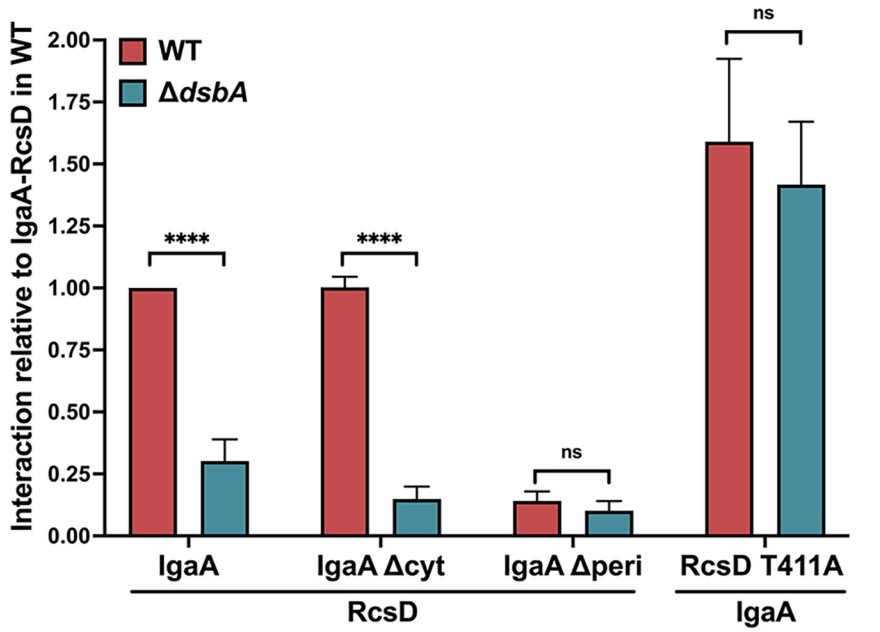

**Fig 4. DsbA is needed for efficient IgaA-RcsD periplasmic interactions. A. IgaA-RcsD interactions are weakened in ΔdsbA.** Beta-galactosidase activity was measured in a *cyaA* mutant strain (BTH101 or AP 58 (BTH101 Δ*dsbA*)) expressing two plasmids encoding the T18 and T25 domains of adenylate cyclase fused to the proteins of interest and the expression measured compared to vector background. The IgaA/RcsD protein fusion plasmids paired with their cognate vector had very low activity; the three controls were averaged and used as "background" for normalization. The IgaA-RcsD interaction was normalized to 1 and other interactions were plotted relative to this interaction in WT. In both Fig 4A and B, the interaction of IgaA-RcsD was 577 units, while the vector control was 29; these units are 1000x the slope of OD420 (see Materials and Methods). This data is compiled from separate sets of assays, each normalized relative to the IgaA/RcsD signal in that experiment. The P values are relative to the *dsbA*⁺ values. **B. Periplasmic interactions are affected in ΔdsbA and a T411A mutation strengthens this IgaA interaction.** IgaA and RcsD/T411A were fused to the T18 and T25 domains, respectively. The IgaA-RcsD interaction in WT was normalized to 1 and all other interactions are plotted relative to this interaction. The RcsD T411A mutation tightens the IgaA-RcsD interaction in WT as well as Δ*dsbA*. The plasmids used were pEAW1 (IgaA-T18), pEAW8 (RcsD-T25), pEAW2 (IgaA-T25), pEAW7 (RcsD-T18), pAP101 (IgaA Δcyt), pEAW1peri (IgaA Δperi), and pEAW8T (RcsD T411A-T25). Values (mean ± SD) from independent experiments were statistically analyzed using multiple unpaired *t*-tests. Statistical significance is shown as follows: ns ($P > 0.05$; non- significant), *** ($P \leq 0.001$), and **** ($P \leq 0.0001$).

The prediction from these interaction assays is that mutations of the IgaA cys residues should also lead to increased activation of the Rcs phosphorelay, as seen for Δ*dsbA* in the presence of DTT. The chromosomal copy of *igaA* was replaced with *igaA*C4S in an *rcsD* mutant strain carrying the P*rprA*::mCherry reporter. While this strain (AP168) is not fluorescent and is non-mucoid, due to lack of RcsD, when a plasmid carrying RcsD was introduced into the cells, the transformed colonies became mucoid, consistent with activation of the Rcs phosphorelay. Growth was poor and it seems likely that this C4S strain, like other strains that express the Rcs phosphorelay at high levels, rapidly picks up suppressing mutations. Therefore, we avoided purification or other manipulation of the transformed colonies. The transformed colonies were grown for assays without further purification in LB and the P*rprA*::mCherry signal assayed; the expression of the reporter was higher than that seen with the Δ*dsbA* strain and was not affected by DTT (S4E Fig). Thus, blocking disulfide bond formation in IgaA via mutation of the cysteine residues induces Rcs expression and mimics the phenotype of the Δ*dsbA* strain in the presence of DTT. Because either pair of cysteine mutants abrogates the full interaction with RcsD (S4C Fig), it seems likely both disulfide bonds are important for proper IgaA folding.

## DjlA can act as a cochaperone and mediate IgaA-RcsD cytoplasmic interactions

DjlA is the third member of the DnaJ domain family in *E. coli*. Proteins with this domain, including DnaJ and CbpA in *E. coli*, collaborate as part of the widely conserved DnaK/Hsp70 chaperone network. DjlA is localized at the cytoplasmic membrane with a short N-terminal region in the periplasm followed by a single transmembrane helix [48]. The conserved J-domain, needed for DnaK interaction, is in the cytoplasm at the C-terminal. A TerB-like (tellurite resistance) domain of unknown function lies between the J-domain and the membrane helix. The role played by this domain has not been clearly defined but it has been speculated to be involved in membrane/lipid-interaction, stress-sensing, DNA processing, and phage defense [49,50]. Although the precise cellular function and substrates of DjlA are unknown, it is a bonafide DnaK cochaperone capable of assisting DnaK in protein refolding [51]. DjlA is not essential for growth and a null mutant did not reveal any significant phenotype except a delayed onset of mucoidy at low temperature [24]. On the other hand, overproduction of DjlA was clearly toxic, with the cells exhibiting growth defects suggestive of a role of DjlA in cell division or membrane integrity [48]. Furthermore, strains overexpressing DjlA demonstrated increased capsule synthesis, the result of Rcs activation, and hypersensitivity to certain drugs

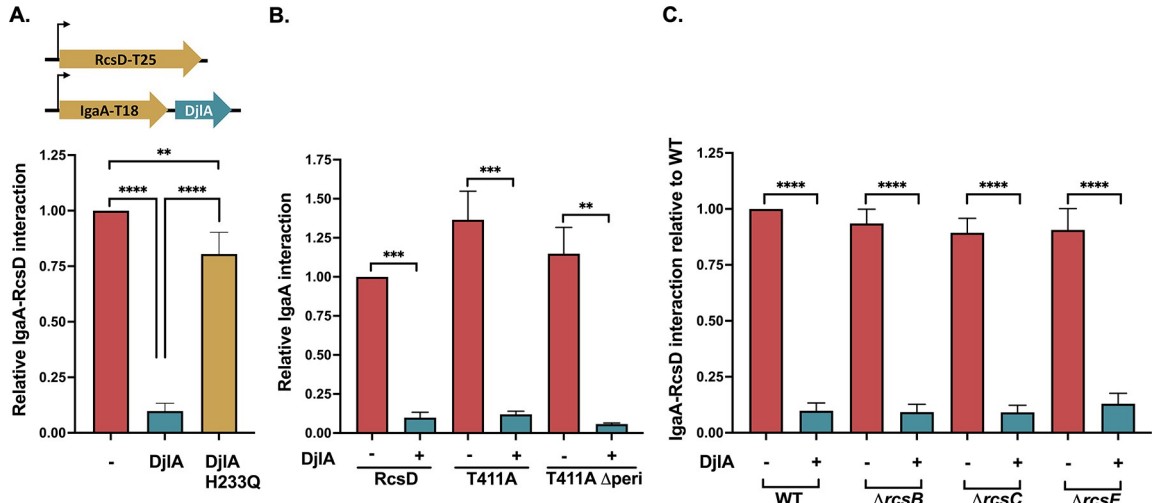

**Fig 5. DjlA weakens IgaA-RcsD interactions. A. DjlA loosens IgaA-RcsD interactions:** Beta-galactosidase activity was measured in a *cyaA* mutant strain (BTH101) in the presence of the indicated plasmids. IgaA and RcsD/RcsD T411A were fused to the T18 and T25 domains, respectively. DjlA or the H233Q mutant of DjlA, inactive as a co-chaperone, was cloned downstream of IgaA under the same promoter control. The IgaA-RcsD interaction was normalized to 1 and all other interactions are plotted relative to this interaction. In both Fig 5A and B, the interaction of IgaA-RcsD was 545 units, while the vector control was 18; these units are 1000x the slope of OD420 (see Materials and Methods). **B. DjlA can disrupt cytoplasmic interactions of IgaA and RcsD:** The IgaA-RcsD interaction was normalized to 1 and all other interactions are plotted relative to this interaction. **C. RcsB, RcsC, RcsF are not needed for DjlA to weaken interactions.** The IgaA-RcsD interaction was tested in BTH101 (WT), EAW 1 (*BTH 101 rcsB::tet*), EAW2 (*BTH 101 rcsC::tet*) and EAW4 (*BTH 101 rcsF::cat*). Plasmids used in this set of experiments were pEAW1 (IgaA-T18), pEAW8 (RcsD-T25), pEAW8T (RcsD T411A-T25), pAP804 (RcsD T411A Δperi -T25), pAP1401 (IgaA-T18 + DjlA), and pAP1402 (IgaA-T18 + DjlA H233Q). The WT interaction here was 533 units, with the vector control as 22 units. Statistical significance was calculated using multiple unpaired *t*-tests and is shown as follows: ** ($P \leq 0.01$), *** ($P \leq 0.001$), and **** ($P \leq 0.0001$).

and chelators [52]. The J-domain is not strictly required for the increased drug sensitivity; however, it is essential for its role as a cochaperone and Rcs activator [24]. DjlA-dependent activation of the Rcs phosphorelay requires DnaK and the nucleotide release factor GrpE, but not DnaJ or CbpA [24]. A functional J-domain and the transmembrane helix were both reported to be essential for Rcs induction by DjlA [24,53,54].

As a membrane-anchored cochaperone, it is possible that DjlA directs DnaK to modulate the folding and/or conformation of an Rcs component or the interaction of Rcs components. This is further supported by our observation that DjlA, uniquely, can overcome the tight interaction between IgaA and RcsD T411A (Fig 3B). Therefore, we tested the effect of DjlA overproduction on the IgaA-RcsD interaction in BACTH assays. We examined the IgaA-RcsD interaction when *djlA* was cloned downstream of IgaA-T18, such that its expression is controlled by the same IPTG-inducible promoter as IgaA. Fig 5A shows that there is a marked decrease in the IgaA-RcsD interactions when DjlA is overexpressed. To confirm that this effect is due to the cochaperone activity of DjlA, we also tested an inactive DjlA mutant, a H233Q mutation known to be defective for DnaK cochaperone activity [24,51]. The overexpression of this DjlA H233Q mutant did not disrupt the IgaA-RcsD interaction. Next, we asked if DjlA can weaken the tight interaction between IgaA and the RcsD T411A mutant (Fig 5B). DjlA overcame this strong interaction and could act even when the periplasmic domain of RcsD was not present (Fig 5B, T411A Δperi). This contrasts with loss of DsbA, which affected the periplasmic but not the cytoplasmic interactions between IgaA and RcsD (Fig 4), and is consistent with the ability of overproduced DjlA to cause Rcs induction even when RcsDT411A is present (Fig 3B). We also tested if any of the other Rcs components are required for DjlA to affect these interactions by carrying out the same assay in derivatives of the BTH101 strain

carrying Δ*rcsB* (EAW1), Δ*rcsC* (EAW2), or Δ*rcsF* (EAW4) mutants. DjlA was able to weaken the interaction between IgaA and RcsD in the absence of each of these proteins (Fig 5C). This suggests that DjlA acts directly as a cochaperone, remodeling the RcsD-IgaA interaction with one or both proteins as direct targets for DjlA.

Previous studies speculated that RcsC is the target of DjlA, likely via transmembrane domain interactions, although that work was done before RcsD and IgaA were known or characterized [53]. As shown above, the periplasmic domain of RcsC was not needed for DjlA induction, but the role of the RcsC transmembrane helices had not been tested. We had previously demonstrated Rcs activation in cells expressing a RcsC_MalF strain containing the RcsC cytoplasmic domain fused to the first two transmembrane (TM) helices of the maltose-transporter membrane protein MalF [18]. RcsC_MalF increased the basal Rcs signal of the strain, but it responded to a RcsF-dependent signal like PMBN [18]. DjlA overexpression also activated Rcs signaling in this strain, demonstrating that DjlA does not need to interact with the RcsC TM helices (S5A Fig).

We further examined the role of the DjlA TM helix in DjlA-dependent Rcs activation. Point mutants in the TM helix of DjlA have been shown to completely block Rcs activation without affecting its stability or localization [53]. Furthermore, the DjlA TM helix is a dimerization domain and contains conserved glycine residues important for Rcs induction [55]. We replaced the DjlA TM helix with the first two MalF TM helices such that the cytoplasmic portion of DjlA is anchored to the membrane with the MalF helices (DjlA_MalF; S5B Fig). We tested P_*rprA*::mCherry induction upon overexpression of this DjlA_MalF variant in a Δ*djlA* Δ*rcsF* strain. We observed that this chimeric DjlA_MalF variant is capable of inducing Rcs activity, unlike the DjlA H233Q mutant or the DjlA ΔTM variant lacking any TM helix (S5B Fig). These results lead us to conclude that the DjlA TM helix does not have an active role in Rcs induction, but that membrane localization is indeed essential. Previous reports did not observe any Rcs induction with either DjlA ΔTM or a MalF-DjlA chimeric variant [24,53,55]. The discrepancy could be due to the somewhat different DjlA_MalF protein construct used by us or may reflect the increased sensitivity of our P_*rprA*::mCherry fluorescent reporter as compared to the *cps-lacZ* beta-galactosidase assay used previously. We further tested the ability of DjlA ΔTM and DjlA_MalF to break the IgaA-RcsD interaction using the BACTH assay. S5C Fig shows that both of these DjlA constructs can disrupt the interaction. This is consistent with the DjlA TM helix not having a specific role in recognizing its Rcs target protein. T18-IgaA levels did not decrease upon overexpression of DjlA (S5D Fig). Surprisingly, DjlA ΔTM was also capable of disrupting the IgaA-RcsD interactions in BACTH even though it was not functional in Rcs activation. Possibly this is due to the difference in protein levels of IgaA/RcsD/DjlA between these two assays. Thus, the TM helix of DjlA is important for inducing activity but may be less important for recognizing its Rcs substrate. We note that others have found that a very similar construct of DjlA lacking the transmembrane region is stable and has co-chaperone function [24,54], consistent with our construct showing function in the BACTH assay.

DjlA was also able to disrupt the interaction of IgaAΔcyt with RcsD (S5C Fig). We have previously shown that the periplasmic interaction of IgaA and RcsD drives the signal in the BACTH assay [18], consistent with the high interaction seen in S5C Fig for IgaAΔcyt with RcsD. Assuming DjlA, with the J-domain in the cytoplasm, interacts with cytoplasmic domains of its substrates, the observation that full-length DjlA can disrupt IgaAΔcyt-RcsD interactions would suggest that DjlA acts on the PAS-like domain of RcsD, leading to conformational changes that not only overcome the T411A interaction with IgaA but also lessen the strong periplasmic contacts between RcsD and IgaA. Interestingly, the DjlA ΔTM was less effective than full length DjlA in disrupting IgaAΔcyt-RcsD interactions.

If multicopy DjlA weakens IgaA-RcsD interactions, it could be speculated that it has a role in modulating these interactions even in single copy under appropriate physiological conditions. However, we failed to find a single copy phenotype of ΔdjlA for Rcs signaling. A strain devoid of DjlA responded like a wild-type strain to varying doses of PMBN (S6A Fig) and responded to A22 and mecillinam (S6B Fig). It has been shown that deletion of djlA rendered a dnaJ deletion strain unable to grow at 40˚ and above [54], suggesting a possible role for DjlA at high temperatures. However, even at 42˚C, no effect of deleting djlA on Rcs basal or induced activity was detected (S6C Fig). While we were unable to detect a role for single copy DjlA for Rcs induction, BACTH assays of IgaA-RcsD interactions showed a somewhat better interaction in the absence of DjlA, consistent with its proposed role in chaperoning this interaction (S6D Fig). Consistent with a role for DjlA for the cytoplasmic interaction of these proteins, an increased interaction of RcsDΔperi with IgaA was also seen in the ΔdjlA strain. RcsD T411A, with a tighter cytoplasmic interaction with IgaA, was unaffected by the absence of DjlA. Based on ribosome profiling, djlA is expressed at a modest level at 37˚ [56]; little is currently known about expression of this protein under other growth conditions. Therefore, we cannot be sure whether DjlA is significantly expressed under these assay conditions. Overall, these results are consistent with the idea that DjlA plays a role in the cytoplasmic signaling interactions of IgaA and RcsD, but under conditions not yet identified.

## DrpB activates Rcs independently of its cell division role

In contrast with DsbA and DjlA, induction by DrpB overexpression has a requirement for the RcsC periplasmic domain for Rcs activation (Fig 2). This observation strongly suggests the possibility that there is signaling to the Rcs phosphorelay via the RcsC periplasmic domain, akin to that seen for many other sensor histidine kinases (reviewed in [57]). Whether there is a direct interaction between DrpB and RcsC for the signaling to occur or the RcsC periplasmic domain senses a signal generated by DrpB overexpression is not yet known. DrpB is a small (100 aa) protein with two transmembrane helices and a small periplasmic region (see schematic in S7A Fig). There are no additional recognizable domains. However, DrpB was identified as a multicopy suppressor of the growth defect of a ΔftsEX mutant at low ionic strength [58]. FtsEX is an ABC transporter that coordinates peptidoglycan remodeling and cell division, allowing cell separation. FtsEX helps in recruitment of downstream cell division proteins to the Z-ring, cell constriction, and activation of periplasmic peptidoglycan amidases [59–61]. FtsEX mutants display chaining defects and have low viability in low-osmolarity medium. DrpB localizes to the divisome and overexpression allowed the growth of an ΔftsEX strain at low osmolarity [58]. Activation of amidases by FtsEX is not strictly required for survival in low salt medium, so it is likely that overproduction of DrpB rescued the growth defect by improving recruitment of downstream components of the divisome [58,61]. Although no cell division or cell morphology defects were observed in a ΔdrpB strain, a double deletion strain with another cell-division protein, DedD, was filamentous and DrpB was found to interact with multiple divisome-associated proteins [58].

RcsB positively regulates a promoter of ftsZ, and multicopy RcsF is a known suppressor of the temperature sensitive ftsZ84 mutant [10]. Although DrpB did not rescue the ftsZ(Ts) phenotype [58], it seemed possible that the ability of DrpB to act as a weak suppressor of the ftsEX mutant is due to its role as an Rcs activator. To test this hypothesis, we asked whether RcsB was needed for the ability of DrpB to suppress the growth defect of ΔftsEX cells in the absence of salt. We overexpressed DrpB from a plasmid-encoded arabinose-inducible promoter in ΔftsEX and ΔftsEX ΔrcsB strains and assayed colony formation on LB medium with and without NaCl. As seen in Fig 6A, multicopy DrpB can rescue the growth defect of ftsEX mutants

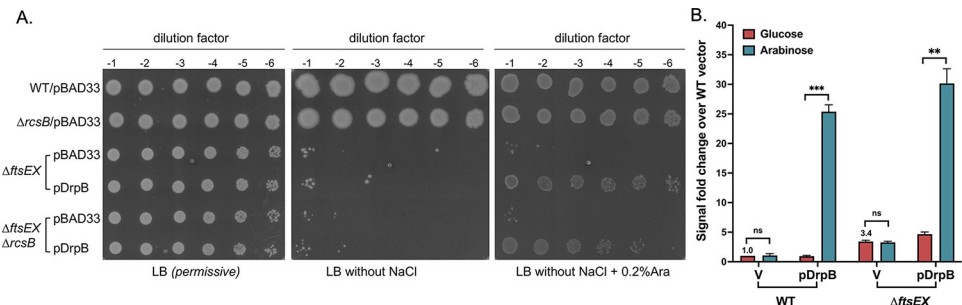

**Fig 6. DrpB signaling is independent of *ftsEX*. A. Role of DrpB as a *ftsEX* suppressor is independent of its role as an Rcs activator:** Strains transformed with the pBAD33 vector or pDrpB (pDSW1977) were grown overnight in LB Miller media with chloramphenicol at 37°C. The cultures were diluted to an $OD_{600}$ of 1 and 4 ul dilutions were spotted on LB Miller (permissive), LB without NaCl, and LB with 0.2% arabinose with no NaCl. All plates contained chloramphenicol (25 μg/ml). Plates were imaged after 16h incubation at 37°C. The strains used are: WT (EC251), *rcsB::kan* (AP158, Δ*rcsB*), Δ*ftsEX* (EC1215), and Δ*ftsEX rcsB::kan* (AP159, Δ*ftsEX* Δ*rcsB*). **B. DrpB can activate Rcs signaling in Δ*ftsEX*:** For the $P_{rprA}$::mCherry assay, the WT (EAW8) and *ftsE::kan* (AP154, Δ*ftsEX*) strains transformed with the pBAD33 vector or pDrpB (pDSW1977) were grown in MOPS minimal glycerol medium (with 150 mM NaCl) containing chloramphenicol and either 0.2% glucose or 0.02% arabinose at 37°C. The RFU at OD 0.4 is plotted relative to uninduced WT with the vector, set to 1. Statistical significance was calculated using multiple unpaired *t*-tests and is shown as follows: ns ($P > 0.05$; non- significant),** ($P \leq 0.01$), *** ($P \leq 0.001$).

even in cells deleted for *rcsB*, demonstrating that the suppression of Δ*ftsEX* is not via Rcs activation. We also asked whether activation of Rcs by DrpB required *ftsEX*, using the $P_{rprA}$::mCherry reporter in a Δ*ftsEX* strain. We found that DrpB indeed induces Rcs in this strain (Fig 6B). Therefore, the role of DrpB in cell growth in the absence of Δ*ftsEX* appears to be independent of Rcs activation and vice versa.

A set of *drpB* mutants were constructed in the pBAD plasmid and tested both for Rcs induction and for ability to rescue the growth of Δ*ftsEX* in the absence of salt (S7 Fig). Since the periplasmic domain of RcsC is needed for DrpB signaling, it could be hypothesized that there is a periplasmic interaction between these proteins. We constructed a DrpB mutant with a deletion of its periplasmic region (Δperi$_{48-57}$); this deletion had no effect on the ability of DrpB to activate Rcs (S7A Fig). However, DrpB Δperi$_{48-57}$ did not rescue the growth of the Δ*ftsEX* mutant, suggesting it is important for this function (S7B Fig). An interaction between the transmembrane regions of DrpB and RcsC also seemed possible. Based on the sequence similarity with other DrpB homologs, we generated point mutants in four highly conserved amino acid residues in and around each of the TM helices of DrpB. S7A Fig shows the effect of these mutants on Rcs activity in a strain deleted for RcsF. Only DrpB T89A had lost the ability to activate Rcs. This allele only slightly decreased suppression of Δ*ftsEX* (S7B Fig). The DrpB R38F had decreased but not absent activity for both induction of Rcs and suppression of Δ*ftsEX* (S7B Fig). We cannot rule out that these mutants are less active due to mislocalization or decreased stability. However, overall, the results confirm the existence of two separable functions of DrpB, with the periplasmic region critical for suppression of Δ*ftsEX* but not for Rcs induction, and T89 critical for Rcs induction but less important to suppress Δ*ftsEX*.

The BACTH assay was used to test if DrpB directly interacts with any of the Rcs components. However, full-length RcsC has previously been found to be non-functional in the BACTH assay [18], and was also negative for interaction with DrpB (S8A Fig). We observed variable and weak (not significant) interactions of DrpB with both IgaA and RcsD (S8A Fig). In a previous study, DrpB interacted with multiple divisome proteins as well as with MalF (used as a control), suggesting the possibility of false-positives in BACTH assays of DrpB with other membrane proteins [58]

As with DjlA, we were not successful in defining an Rcs phenotype for deletion of *drpB*. The Δ*drpB* strain responds to PMBN, A22 and mecillinam (S8B Fig), which is consistent with a previous observation that DrpB is not needed for the Rcs response to osmotic shock or ethanol stress [28]. Multicopy DrpB also was previously found to induce the *psp* (phage shock) signaling pathway [28]. Rcs induction by DrpB was unaffected by the absence of *psp* components *pspC* and *pspF* (S8C Fig), ruling out induction of Rcs indirectly via induction of Psp. Ribosome profiling suggests a modest expression of DrpB from the chromosome [56], but very little is known about the roles and expression level of the single copy DrpB protein. More studies are needed to ascertain if multicopy DrpB activates the Rcs cascade by interacting directly with RcsC or more indirectly. More broadly, we do not know whether there is a physiological condition during which DrpB made from the single copy gene acts to promote Rcs activation, or whether the multicopy effect triggers a stress that leads to activation, but we would expect a critical role for the RcsC periplasmic domain in either case.

Similar to DrpB, a group of small cytosolic proteins in the *ycgz-ymgABC* operon, which is involved in biofilm formation and acid resistance, have been demonstrated to trigger the Rcs cascade [30,62]. AriR/YmgB, in complex with YcgZ and YmgA, was reported to target RcsC to induce the Rcs pathway [63]. We tested the Rcs activation by YmgB in our strains and found it to be RcsF independent and blocked by the RcsD T411A mutation (S9 Fig). However, in contrast with DrpB, it is independent of the periplasmic domain of RcsC, indicating a different mode of Rcs activation.

As noted above, the BarA phosphorelay protein was shown to regulate RcsB in *Salmonella*, independent of RcsF [26]. While our identification of the mode of action of DsbA, DrpB and DjlA would suggest that they act upstream of RcsB, we tested whether BarA was necessary for these factors to induce Rcs. First, the effect of a *barA* mutant on expression of our reporter under normal growth was tested (S10A Fig). The basal expression of the reporter was modestly reduced in the *barA* mutant, suggesting that, as in *Salmonella*, BarA may signal to regulate RcsB; a reduction was also seen in strains deleted for *rcsC* or *rcsD*, but not for the *rcsB* deletion, consistent with an effect on RcsB. Overexpression of DjlA or DrpB were each still able to induce the reporter in a *barA* mutant (S10B Fig, compare to Fig 1B). Similarly, deletion of *dsbA* induced signaling in the absence of *barA* (S10C Fig). Therefore, these three RcsF-independent mechanisms are also independent of BarA.

## Discussion

The Rcs phosphorelay is a remarkably sophisticated signaling pathway capable of responding to a wide array of stress signals at the cell envelope. RcsF plays a vital role in sensing most of the Rcs induction cues but is not strictly necessary for activating signaling through the Rcs phosphorelay. Our study elucidates diverse modes of activation of the Rcs phosphorelay which are independent of RcsF (Fig 7). The results demonstrate that RcsF is not the sole sensor for this pathway. In particular, our identification of DrpB as an RcsF-independent activator suggests the possibility of other yet unidentified RcsF-independent activators that act via the RcsC periplasmic domain.

We delineate how three RcsF-independent activators trigger the Rcs phosphorelay, each in a different fashion. Our current understanding of the regulation of Rcs signaling is primarily based on the interactions of IgaA with RcsF and RcsD [2,3,16–18,22]. Interactions of RcsF with IgaA form the first level of regulation crucial for activation of signaling (Fig 1A). Signals detected by RcsF change the interaction of IgaA with RcsD; this IgaA-RcsD interaction serves as the regulatory switch necessary for signal activation. Dependent on the nature of these interactions, the RcsC histidine kinase acts as either a phosphatase (RcsB response regulator not phosphorylated, inactive) or a kinase (RcsB response regulator phosphorylated and active).

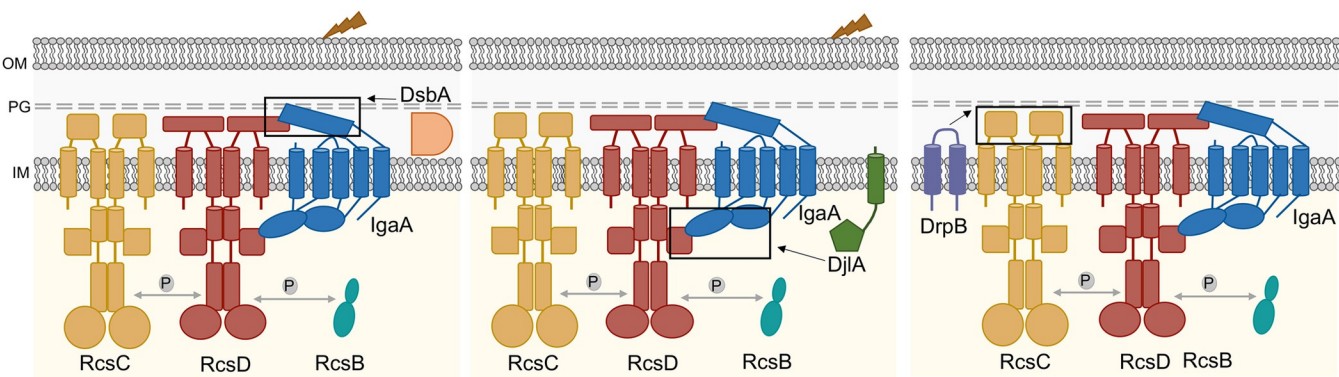

**Fig 7. Diverse modes of RcsF-independent activation by DsbA, DjlA, and DrpB.** First, *dsbA* mutants activate the Rcs phosphorelay in response to DTT and PMBN. The absence of DsbA likely leads to defective disulfide bond formation and misfolding of the IgaA periplasmic domain. Thus, DsbA is needed to maintain proper IgaA-RcsD interactions in the periplasm and in its absence, IgaA cannot effectively regulate Rcs signaling. Second, DjlA acts as a cochaperone for IgaA-RcsD interactions. Overexpression of DjlA weakens the IgaA-RcsD interactions in the cytoplasm leading to Rcs activation. Third, DrpB overexpression induces the Rcs cascade but it does not act directly on IgaA-RcsD interactions like DsbA and DjlA. Uniquely, DrpB requires the RcsC periplasmic domain to activate Rcs signaling and either activates it by direct interaction with RcsC or via an indirect mechanism involving RcsC as the sensor instead of RcsF.

The IgaA-RcsD interactions are also essential for repression of signaling, thereby preventing constitutive activation and the resulting lethality. Our work underscores the importance of these IgaA-RcsD interactions in regulating this cascade even in the absence of RcsF-dependent activation. Activation in all three cases, in cells deleted for DsbA or overproducing DjlA or DrpB, is dependent on RcsC and RcsD and does not initiate activation merely by increasing expression or phosphorylation of RcsB.

## Interrupting the interactions of RcsD and IgaA induce Rcs signalling

Loss of DsbA or DjlA overproduction each activate the phosphorelay by weakening the interactions between IgaA and RcsD, affecting their folding/conformation via distinct modes. We have previously shown that IgaA and RcsD interact both through their periplasmic domains and their cytoplasmic domains ([18], Fig 1A). The periplasmic strong contact provides an anchor that keeps signaling from rising too high. Based on the identification of the RcsDT411A mutant in the cytoplasmic PAS domain of RcsD that blocks RcsF-dependent induction, the cytoplasmic contact was defined as the regulatory switch region [18]. The loss of DsbA impacts the periplasmic contacts, likely due to its role in formation of disulfide bonds in IgaA, and therefore IgaA folding. This is reflected in a decrease in the interaction of IgaA and RcsD in the absence of DsbA and in elevated Rcs activation, further increased when cells are treated with DTT. Presumably in a strain devoid of DsbA, a fraction of IgaA is in a reduced state with one or both disulfide bonds disrupted; addition of DTT leads to the disruption of any remaining disulfide bonds. Mutating the IgaA periplasmic cysteines predicted to be involved in these disulfide bonds mimics the *dsbA* mutant, decreasing the interaction of IgaA and RcsD (S4C Fig) and increasing signaling (Fig 1C). RcsF itself requires disulfide bonds for proper folding and function [7,44]. Therefore, it is not surprising that *dsbA* mutants no longer recognize RcsF-dependent cues such as A22 and mecillinam (S4A Fig), since RcsF is non-functional in this strain. Similar observations have been made in *Salmonella*; *Salmonella dsbA* mutants activate Rcs, independent of RcsF, and this is increased upon DTT addition, while a strain carrying a *bamB* deletion induces Rcs, dependent upon RcsF and is unaffected by DTT treatment [64].

We might have also expected PMBN, which normally induces the Rcs system in an RcsF-dependent fashion (see Fig 1C) to be ineffective in the *dsbA* mutant. However, it induces in

much the manner that DTT does, independently of *rcsF* (Fig 1C). This result suggests that PMBN causes some redox stress, in addition to the LPS/membrane damage that leads to RcsF-dependent induction. When the Dsb network is functional, DTT cannot induce Rcs activation and any PMBN induction requires RcsF, suggesting that Rcs probably does not monitor disulfide bond formation in the periplasm as part of its normal function. Instead, the disulfide bonds are necessary for proper folding of IgaA to allow the periplasmic interaction with RcsD. We cannot rule out the possibility that growth conditions exist under which IgaA disulfide bonds might be disrupted sufficiently to induce Rcs signaling. A study of the roles of the periplasmic cysteines of IgaA in *Salmonella* also found that disulfide bonds were critical both for proper repression of Rcs and, to some extent, for viability of the cells [45].

Overproduction of DjlA, a membrane-localized DnaJ family protein, also loosens the interaction of IgaA and RcsD (Fig 5A) and turns up phosphorelay activity independently of RcsF (Fig 1B); these effects depend on an active J domain (mutated in DjlA H233Q). However, the primary effect of DjlA overexpression is likely on the cytoplasmic domains of RcsD, IgaA, or both. This is suggested both by the observation that DjlA overproduction but not loss of DsbA overcomes the RcsD T411A mutation (Fig 3) and by the location of the J domain in the cytoplasm. The ability of DjlA overproduction to induce signaling also allows us to conclude that the interaction of the RcsD PAS-like domain with the IgaA cytoplasmic domain can regulate activation by both RcsF-dependent and RcsF-independent signals.

DjlA was the only RcsF-independent activator which overcame the heightened interaction between RcsD T411A mutant and IgaA. This would be consistent with DjlA acting as a chaperone to mediate these interactions. Molecular chaperone systems can assist protein-protein interactions by loosening and altering protein conformations (reviewed in [65]). Unlike DnaJ and CbpA, the other *E. coli* DnaJ family proteins capable of working with DnaK, DjlA does not carry a specific substrate-binding domain, so it is not known how it identifies its clients [53]. Since only DjlA, and not the other two DnaJ family proteins, induces Rcs, it seemed possible that the membrane insertion of DjlA is critical in recruiting DnaK to the sites of IgaA/RcsD, reinforced by the lack of Rcs activation from a cytosolic soluble form of DjlA ([24,53], S5B Fig). However, we found that DjlA fused to the MalF transmembrane domains can activate Rcs, strongly suggesting this helix is not required for Rcs interaction. Our results are most consistent with RcsD as the target of DjlA function, based on the ability of DjlA to disrupt interaction of IgaAΔcyt and RcsD (S5C Fig). Because the periplasmic interaction of RcsD and IgaA is sufficient for a strong interaction signal in the BACTH assay, we can also conclude that the action of DjlA on the cytoplasmic domain of RcsD must lead to a change in the RcsD periplasmic domain.

The effects here were seen when DjlA was overproduced. We do not know whether or when the chromosomally encoded DjlA is able to modulate the IgaA-RcsD interaction. Deletion of *djlA* strengthened the interaction of IgaA and RcsD in the BACTH assay (S6D Fig) but did not have any effect on Rcs induction under a variety of conditions (S6 Fig). Not much is known about the physiological signals regulating DjlA expression; DjlA-stimulation of Rcs might be relevant under specific growth conditions or might be important for particular inducing signals not tested here. There is also evidence that DjlA acts as a negative regulator of Rcs [12,66]. Those studies demonstrated a stimulation of the Rcs activation upon deletion of DjlA, particularly in mutant strains such as *mdo* and *rfa* which are already activated for Rcs in an RcsF-dependent manner. Lack of DjlA could increase the envelope stress caused by mutants in *mdo* and *rfa* independently of the RcsF-independent effect seen here. Undoubtedly, DjlA has substrates other than the Rcs system, and possibly the opposing effects (controlling envelope stress to negatively regulate Rcs and overcoming the RcsD/IgaA protein to induce Rcs) balance each other, making it more difficult to define a single copy effect.

### Rcs activation via histidine kinase RcsC

Uniquely among the three proteins we investigated, DrpB required the RcsC periplasmic domain for Rcs activation. RcsC is a member of a broad family of histidine kinases; in many of these, the periplasmic domains are critical for sensing environmental signals and regulating histidine kinase activity [67–69]. In our previous work, we found that IgaA interacted with RcsD, the phosphorelay protein, rather than RcsC and Rcs inducing treatments were independent of the RcsC periplasmic domain [18]. While RcsC activity (kinase vs. phosphatase) must be adjusted by RcsF-dependent induction, there was no previous evidence for a role of RscC in signal sensing. Thus, the identification of DrpB and its dependence on the RcsC periplasmic region demonstrates the likelihood that some signals likely act in the "classical" way seen for other histidine kinases; if so, such signals would be independent of RcsF. While we do not know if DrpB acts directly or indirectly on RcsC, it is tempting to speculate that the RcsC periplasmic domain is sensing a signal created by DrpB overexpression, allowing RcsC to activate signaling. Interestingly, a ribosome profiling study revealed DrpB as one of the proteins upregulated during severe acid stress in *E. coli* and this may be linked to its Rcs function [70]. Alternatively, the overproduction of DrpB itself is a source of stress which is detected by RcsC. DrpB overproduction also leads to the induction of another envelope stress response, the Psp response pathway [28]. The Psp response primarily detects and mitigates stress at the inner membrane (reviewed in [71]). While we find that the Psp pathway is not necessary for Rcs induction by DrpB (S8 Fig), it will be of interest to see whether other inducing signals shared by these two systems also share dependence upon the RcsC periplasmic domain.

Further evidence for RcsC as a sensor comes from studies on the protein YmgB. YmgB is a cytoplasmic protein involved in biofilm formation and acid-resistance and it stimulates the Rcs pathway, reportedly by interacting with the RcsC cytoplasmic domain [62,63]. In our hands, and consistent with the model suggested by DrpB overexpression, YmgB-dependent induction is RcsF-independent, but, unlike DrpB, is also independent of the periplasmic domain of RcsC (S9 Fig). This is consistent with previous work on YmgB and raises the possibility that RcsC in fact senses different signals with its cytoplasmic and periplasmic domains.

A number of other proteins have been found to affect Rcs activity. BarA plays a role in the induction of the Rcs system in *Salmonella*, via effects on RcsB synthesis [26]. YpdI, a lipoprotein of unknown function, induces mucoidy in the absence of RcsF [29]. Activation of Rcs in a double mutant of *yqjA* and *yghB*, both membrane proteins and putative lipid flippases, is partially dependent on RcsF [33,72]. Deciphering their mechanisms could add to our knowledge about Rcs regulation.

Overall, our results emphasize the versatility and flexibility of the Rcs pathway. RcsF senses stress at the outer cell surface and in the periplasm. The RcsF-independent signaling pathways studied here suggest that, in addition, RcsC may play a role in sensing stressors at the cytoplasmic membrane, and possibly in the cytoplasm as well. Possibly this sensing by RcsC represents the remnants of the original version of this system, before RcsD, IgaA and RcsF evolved to sense a broader set of inducing signals. Clearly there is still much to be learned about this system, across species, and in different natural environments. We look forward to better understanding this unique system in the future.

## Materials and methods

### Bacterial growth conditions and strain construction

The strains were grown in LB medium with appropriate antibiotics (ampicillin 100 µg/ml, kanamycin 30–50 µg/ml, chloramphenicol 10 µg/ml for *cat-sacB* allele or 25 µg/ml for chl[R]

plasmids, tetracycline 25µg/ml, zeocin 50µg/ml). Glucose at a final concentration of 1% was added to reduce the basal expression from the plasmids containing pBAD and pLac promoters.

Strains, plasmids, primers and synthetic DNA fragments (gBlocks) used in this study are listed in S1–S4 Tables respectively. Oligonucleotides and gBlocks were from IDT DNA, Coralville, IA. Strains were constructed by recombineering or P1 transduction with selectable markers (S1 Table). Recombineering was done in strains carrying a chromosomal mini-λ Red system (*miniλ::tet*) or a plasmid-borne Red system (pSIM27). Some strains were generated by direct P1 transduction from the corresponding mutant strains in the Keio collection [73]. Plasmids were constructed by the Gibson assembly method using the In-fusion HD Cloning kit (Takara Bio USA) [74]. PCR products were purified using column purification (Qiagen) and transformed into NEB DH5α F'*lacIQ* cells. Site-directed mutagenesis in the genes was carried out using the QuikChange Site-directed mutagenesis kit (Agilent). Sequencing was done to confirm the chromosomal and plasmid modifications.

## Screening for multicopy RcsF-independent activators

A pBR-based plasmid library [75] was transformed into EAW34 electrocompetent cells. Aliquots of the transformed mixture were plated on LB plates containing 100 µg/ml ampicillin and incubated overnight at 37°C. Subsequently, the plates were imaged and the colonies displaying high fluorescence were restreaked on LB ampicillin plates and purified. The plasmid DNA was isolated from these colonies and retransformed into EAW34 to validate their phenotype. The positive candidate plasmids were isolated, sequenced and the reads mapped to the *E. coli* K-12 MG1655 genome to identify the inserts.

## Fluorescence reporter assays

Growth and fluorescence assays for Rcs activation were carried out in 96-well plates in a Tecan Spark microplate reader. All the strains carried a $P_{rprA}$::mCherry transcriptional fusion at the *ara* locus as a reporter for Rcs signaling [18]. For strains carrying overexpression plasmids expressing DjlA, DrpB, or RcsF, the overnight cultures were grown in MOPS minimal glucose (0.2%) medium (Teknova). For the $P_{rprA}$::mCherry assays for these strains, they were diluted to $OD_{600}$ 0.03–0.05 in MOPS minimal glycerol media (0.05% glucose, 0.5% glycerol) with 0.02% arabinose (induced) or 0.2% glucose as an uninduced control. Both the optical density ($OD_{600}$) and mCherry fluorescence were monitored every fifteen minutes for 12 hours. Polymyxin B nonapeptide (PMBN; Sigma) was used at 20 µg/ml, to induce the Rcs system. A22 (an MreB inhibitor) was used at 5 µg/ml, and mecillinam was used at 0.3 µg/ml. Each assay was performed in technical duplicates in the microtiter plate, with the biological replicates performed on different days. The fluorescence values at equivalent $OD_{600}$ values (0.4 ± 0.04) for each strain were converted to bar graphs and plotted as signal fold change of fluorescence signal over the uninduced wild type strain. Error bars represent the standard deviation of at least 3 assays. For strains that did not reach an $OD_{600}$ value of 0.4, the OD is noted on the bar graph. For the $P_{rprA}$::mCherry assays for the strains not carrying any plasmids, MOPS minimal glucose media was used for growing overnight cultures and performing the fluorescence assays. For the assays pertaining to *dsbA* mutants, the overnight cultures were grown in LB medium. Subsequently, cells were washed with and diluted to $OD_{600}$ 0.05 in MOPS minimal glucose media to carry out the $P_{rprA}$::mCherry assay. To activate the Rcs system, Dithiothreitol (DTT, Sigma) was added at a final concentration of 1mM or Polymyxin B nonapeptide (PMBN; Sigma), was used at 20 µg/ml. All assays were performed at 37°C, unless otherwise indicated. Values derived from at least three independent experiments were plotted to show

the mean with error bars indicating standard deviation. These were statistically analyzed by multiple unpaired *t*-tests using GraphPad Prism 10 software. The primary data for all graphs are found in S1 Data.

## Bacterial adenylate cyclase two hybrid assay (BACTH)

For the bacterial adenylate cyclase two hybrid assay (BACTH), an adenylate cyclase mutant strain (BTH101) or a derivative if this strain was used. The proteins whose interactions were being examined were cloned into the T18 and T25 portions of adenylate cyclase [46,47]. The reconstitution of adenylate cyclase upon their interaction allows production of cAMP, which interacts with CRP to activate the *lac* operon, assayed as beta-galactosidase activity. In the absence of interaction, T18 and T25 do not get reconstituted to form a functional adenylate cyclase and the strain does not express beta-galactosidase. The proteins used in this study were fused at their C-terminal to the Cya fragments. Plasmids expressing IgaA or RcsD as T18/T25 fusion constructs were cotransformed into BTH101, plated on LB-agar medium containing 100 μg/ml ampicillin and 50 μg/ml kanamycin and incubated at 30˚C for 2 days. To measure the beta-galactosidase activity, the resulting colonies were inoculated and grown overnight in LB medium containing 100 μg/ml ampicillin, 50 μg/ml kanamycin, and 0.5 mM IPTG at 30˚C. The beta-galactosidase assay was performed in 96-well plates. The activity was analyzed by measuring the kinetics of ONPG degradation by monitoring the OD 420 nm at 28˚C for 40 min at intervals of 1 min in a Tecan Spark microplate reader. The beta-galactosidase activity was calculated using the slope of OD 420 divided by their OD 600. Each protein fusion paired with the cognate vector produces very little activity and these were used as negative controls. Every graph is compiled from at least 3 separate sets of assays, and the beta-galactosidase activity was plotted relative to the IgaA/RcsD interaction in that particular experiment.

To test the effect of DsbA activity on the various IgaA/RcsD constructs, the BACTH assays were carried out in BTH101 and a *dsbA* mutant derivative of BTH101 (AP58). To test the effect of DjlA on the IgaA/RcsD interactions, d*jlA* or mutants of *djlA* were cloned downstream of IgaA under the control of the same promoter. The assay was carried out in BTH101 or its derivatives EAW1, EAW2, and EAW4 as above and interactions plotted relative to the IgaA/RcsD interaction in WT, set to 1. The primary data for all graphs are found in S1 Data.

## Western blots

For S4D Fig, plasmids were transformed into EAW8 and EAW62 and for S5D Fig, the plasmids were transformed into *E. coli* DH5α competent cells (NEB). The colonies were inoculated into LB media containing Amp (100 μg/ml) and 1 mM IPTG and then grown for 8 hours at 30˚C. One ml of culture was precipitated using TCA, washed in acetone and resuspended in LDS sample buffer standardized to OD. 8 μl volumes were loaded on 4–12% NuPage gradient gels (Invitrogen, CA) and then transferred onto nitrocellulose membranes in an iBlot2 transfer device as per manufacturer's specifications (Invitrogen, CA). The membranes were blocked with Casein PBS and then incubated in CyaA-T18 mouse monoclonal primary antibody at 1:10,000 (Santa Cruz Biotechnology, CA) or/and a mouse monoclonal anti-EF-Tu at 1:10,000 (Hycult Biotech, PA) overnight at 4˚C. Secondary fluorescent antibody anti-mouse DyLight800 (Bio-Rad, CA) was used at 1:10,000. Imaging was done on a ChemiDoc MP imaging system (Bio-Rad).

## Viability testing of *ftsEX* mutants

The wildtype and *ftsEX* deletion strains were transformed with the pBAD33 vector or a DrpB containing plasmid and plated on LB Miller-agar medium at 30˚C containing 25 μg/ml

chloramphenicol. Overnight cultures were grown in LB Miller medium 25 μg/ml chloramphenicol at 30°C. Cells were harvested from 1 mL of culture and diluted to $OD_{600}$ 1.0 in LB Miller or LB0N (LB without any NaCl) medium. Serial dilutions (10-fold) were made and 4 μl was spotted onto LB Miller-agar or LB0N-agar plates containing 25 μg/ml chloramphenicol. Arabinose was added at 0.2% for induction in LB0N. Plates were imaged after 16 h of growth at 37°C.

## Supporting information

**S1 Table. List of strains used in this study.**
(DOCX)

**S2 Table. List of plasmids used in this study.**
(DOCX)

**S3 Table. List of primers used in this study.**
(DOCX)

**S4 Table. List of gBlocks used in this study.**
(DOCX)

**S1 Data. Excel file with primary data for all graphs.**
(XLSX)

**S1 Fig. Signaling by putative RcsF-independent Rcs activators.** A genomic library was transformed into an *rcsF* deletion strain (EAW34) containing the $P_{rprA}$::mCherry reporter for Rcs signaling and the colonies were screened for increased fluorescence. The plasmids were isolated from colonies showing high fluorescence (indicating Rcs activation) and sequenced. The genes present in the candidate plasmids, representing two genomic regions, are shown here. **A.** The gene encoding the YedR/DrpB ORF (indicated here in red) was present in 13 out of 14 candidate plasmids, in 8 independent clones with different end points. **B.** The remaining candidate plasmid contained genes for Spot42 sRNA (*spf*) and the YihA ORF. **C.** For this $P_{rprA}$::mCherry assay, the strain AP 51 (*rcsF*::*kan*) overexpressing pBR-plac (V), pBR-plac-*rseX* or pBR-plac-*spf* was grown in MOPS minimal glycerol medium containing 100 μg/ml ampicillin and their fluorescence measured over time at 37°C. The RFU at OD 0.4 is plotted with the value for the vector in the absence of IPTG set to 1. The strains were induced with 100 μM IPTG. **D.** The AP51 strain containing pBAD33 or pBAD33-*yihA* (pAP3327) was grown in MOPS minimal glycerol medium with 25 μg/ml chloramphenicol and induced with 0.02% arabinose. Data from three independent experiments is plotted as mean with error bars indicating the standard deviation. Values were statistically analyzed using multiple unpaired *t*-tests. 'ns' indicates a *P*-value > 0.05 (non- significant).
(TIF)

**S2 Fig. Dependence of DsbA, DjlA, and DrpB on each other for Rcs activation. A. Effect of DjlA or DrpB deletion upon Rcs signaling in *dsbA* mutants:** All the strains carry an *rprA* promoter fusion to mCherry ($P_{rprA}$-mCherry) and the mCherry fluorescence acts as an indicator for Rcs activation. For the $P_{rprA}$::mCherry assay, the cells were grown in MOPS minimal glucose medium at 37°C. The RFU at OD 0.4 for the WT uninduced was set to 1 and the relative induction compared to that depicted here (top panel). The cells were treated at the beginning of growth with either 1mM DTT or 20μg/ml PMBN. Strains used were: WT (EAW8), Δ*dsbA djlA*::*kan* (AP72), and Δ*dsbA drpB*::*kan* (AP71). The middle panel depicts the relative fluorescence units (RFU) as a function of time and the lower panel shows a representative

growth curve for each strain. **B. Signaling upon overexpression of DjlA or DrpB in *dsbA* mutants:** The strains WT (EAW8) and *dsbA*::*kan* (EAW62) carry the P*rprA*::mCherry promoter fusion. The strains containing plasmids overexpressing DjlA (pDjlA/pPSG961) or DrpB (pDrpB/pDSW1977) were grown in MOPS minimal glycerol medium with 25 μg/ml chloramphenicol at 37°C and 0.02% arabinose was added for induction. The RFU at OD 0.4 for the WT vector uninduced was set to one; other results are normalized to that value. The middle and bottom panels depict the RFU and $OD_{600}$ as a function of time. **C. Rcs activation by overexpression of DjlA and DrpB in Δ*drpB* and Δ*djlA* strains:** The strains overexpressing DjlA (pDjlA/pPSG961) or DrpB (pDrpB/pDSW1977) were grown as in B. Strains used were: WT (EAW8), *drpB*::*kan* (AP41), and *djlA*::*kan* (AP46). Data from three independent experiments is plotted as mean with error bars indicating the standard deviation. Values were statistically analyzed using multiple unpaired *t*-tests. Statistical significance is indicated as follows: ns ($P > 0.05$; non- significant), * ($P < 0.05$), ** ($P \leq 0.01$), *** ($P \leq 0.001$).
(TIF)

**S3 Fig. Signaling by multicopy RcsF.** The strains overexpressing RcsF (pAP3340) or its inner membrane localized mutant RcsF$^{IM}$ (pAP3341) were grown in MOPS minimal glycerol medium containing chloramphenicol (25 μg/ml) and either 0.2% glucose or 0.02% arabinose at 37°C. The RFU at OD 0.4 compared to the uninduced vector control, set to 1, was plotted. The strains used were: WT (EAW8), *rcsF*::*kan* (AP51), *rcsC*Δ*peri* (EAW70), and *rcsD T411A* (EAW121).
(TIF)

**S4 Fig. A. Δ*dsbA* does not respond to A22 and Mecillinam.** For the P*rprA*::mCherry assay, the strains were grown in MOPS minimal glucose medium at 37°C. The RFU at OD 0.4 as compared to WT uninduced, set to 1, is depicted here. The cells were subjected to either 5 μg/ml A22 or 0.3 μg/ml mecillinam. The strains used were: WT (EAW8), *rcsF*::*cat* (EAW32), *dsbA*::*kan* (EAW62), and *dsbA*::*kan rcsF*::*cat* (EAW67). **B. Rcs signaling in RcsC cysteine mutant strains.** For the P*rprA*::mCherry assay, the strains were grown in MOPS minimal glucose medium at 37°C. The RFU at OD 0.4 as compared to WT uninduced, set to one, is shown here. The cells were subjected to either 1mM DTT or 20μg/ml PMBN. The *dsbA* RcsC C2A (*rcsC* C111A C154A) mutant responded to DTT and PMBN similarly to the Δ*dsbA* strain. The strains used were: WT (EAW8), *dsbA*::*kan* (EAW62), *rcsC C2A atoS*::*kan* (AP172), and Δ*dsbA rcsC C2A atoS*::*kan* (AP173). **C. Interaction of IgaA Cysteine mutants with RcsD.** Beta-galactosidase activity was measured in bacterial two hybrid assay, in the standard *cyaA* mutant strain (BTH101 or in a strain mutant for both *cyaA* and *dsbA*, AP 58 (BTH101 Δ*dsbA*)). The IgaA-RcsD interaction in WT was normalized to 1 and other interactions plotted relative to this interaction. The interaction of IgaA-RcsD was 569 units, while the vector control was 20; these units are 1000x the slope of OD420 (see Materials and Methods). This data is compiled from separate sets of assays, each normalized relative to the IgaA/RcsD signal in that experiment. The T18 tagged plasmids used are pEAW1 (IgaA-T18), pAP102 (IgaA C404S C425S), pAP103 (IgaA C498S C504S), pEAW1C4S (IgaA C4S) pEAW1peri (IgaA Δperi), pEAW8 (RcsD-T25), and pEAW8T (RcsD T411A-T25). See Fig 4B for IgaAΔperi structure. **D. Expression of IgaA-T18 fusions.** EAW8 (WT) and EAW62 (Δ*dsbA*) expressing the IgaA-T18 fusion constructs were probed with the anti-CyaA antibody and anti-EF-Tu antibody (loading control). Lane 1 shows the protein markers; Lanes 2–5 and Lanes 6–9 show expression in EAW8 and EAW62 respectively. Lanes 2, 6: pEAW1 (IgaA), Lanes 3, 7: pAP101 (IgaA Δcyt), Lanes 4, 8: pEAW1peri (IgaA Δperi), and Lanes 5, 9: pEAW1C4S (IgaA C4S). See Fig 4B for structures of IgaA Δcyt and IgaA Δperi. **E. Signaling by IgaA cysteine mutants.** For the P*rprA*::mCherry assay, the strains EAW19 (Δ*rcsD*), AP168 (*igaA C4S* Δ*rcsD*), and AP169 (Δ*rcsD dsbA*::*kan)*

were transformed with RcsD (pEAW11) and plated on LB Ampicillin plates. The resulting transformed colonies were inoculated without further purification in LB medium with glucose (0.2%) and Ampicillin, grown at 37˚C and their fluorescence monitored. This protocol was used to avoid the accumulation of suppressors seen in the C4S strain during purification and passaging. The cells were subjected to nothing (uninduced) or to 1mM DTT from the beginning. The RFU/OD after 4h of growth is plotted and the OD at that time point is shown for each bar. The growth profile of the strains during the experiment is shown on the right. Statistical significance was calculated using multiple unpaired $t$-tests and is indicated as follows: ns ($P > 0.05$; non- significant), * ($P < 0.05$), ** ($P \leq 0.01$), *** ($P \leq 0.001$), and **** ($P \leq 0.0001$). (TIF)

**S5 Fig. DjlA TM helix does not recognize a Rcs target. A. DjlA signaling in a RcsC$_{MalF}$ strain:** For the P$_{rprA}$::mCherry assay, the strains WT (EAW8) and RcsC$_{MalF}$ (EAW72) overexpressing DjlA (pDjlA/pPSG961) or pBAD33 vector were grown in MOPS minimal glycerol medium containing chloramphenicol (25 µg/ml) and either 0.2% glucose or 0.02% arabinose at 37˚C. The RFU at OD 0.4 uninduced for the WT was set to 1 (first column), and relative values to that are plotted. EAW72 carries a chimeric variant of RcsC which has its cytoplasmic domains fused to the MalF TM helices. Statistical significance was calculated using multiple unpaired $t$-tests and is indicated as follows: ns ($P > 0.05$; non- significant), * ($P < 0.05$), ** ($P \leq 0.01$), *** ($P \leq 0.001$), and **** ($P \leq 0.0001$). **B. Signaling by DjlA mutants:** For the P$_{rprA}$::mCherry assay, the strain AP 163 ($djlA$::$kan$ $\Delta rcsF$) carrying the pBAD33 vector or pBAD33 derivatives expressing DjlA (pDjlA/pPSG961), DjlA H233Q (pAP3315), DjlA ΔTM (pAP3311), or DjlA$_{MalF}$ (pAP3312) were grown as in A. The RFU of the strain carrying the uninduced vector at OD 0.4 was set to 1 and results normalized to that are plotted. **C. The DjlA TM helix is not needed for weakening IgaA-RcsD interactions.** Beta-galactosidase activity was measured in a $cyaA$ mutant strain (BTH101). IgaA and RcsD/RcsD T411A were fused to the T18 and T25 domains, respectively. DjlA or its variants were cloned downstream of IgaA under the same promoter control. IgaA-RcsD interaction in the absence of DjlA was normalized to 1 and all other interactions are plotted relative to this interaction. The interaction of IgaA-RcsD was 580 units, while the vector control was 27; these units are 1000x the slope of OD420 (see Materials and Methods). Plasmids used were pEAW1 (IgaA-T18), pEAW8 (RcsD-T25), pAP101 (IgaA Δcyt–T18), pAP1401 (IgaA-T18 + DjlA), pAP1403 (IgaA-T18 + DjlA ΔTM), and pAP1404 (IgaA-T18 + DjlA$_{MalF}$). **D. Expression of IgaA-T18 fusion.** $E. coli$ DH5α strain expressing the IgaA-T18 fusion constructs were probed with the anti-CyaA antibody and anti-EF-Tu antibody (loading control). Lane 1: protein marker; Lane 2: pEAW1 (IgaA-T18), Lane 3: pAP1401 (IgaA-T18 + DjlA), Lane 4: pAP1403 (IgaA-T18 + DjlA ΔTM). (TIF)

**S6 Fig. Rcs signaling in Δ$djlA$. A. Dose-dependent PMBN response in Δ$djlA$:** For the P$_{rprA}$:: mCherry assay, the strains WT (EAW8) and $djlA$::$kan$ (AP46) were grown in MOPS minimal glucose medium with or without PMBN (5 µg/ml, 10 µg/ml, or 20 µg/ml) at 37˚C. The RFU at OD 0.4 as compared to WT uninduced, set to 1, is depicted here. Values (Mean ± SD) from independent experiments were statistically analyzed using multiple unpaired $t$-tests. The fluorescence signal of each PMBN dose in Δ$djlA$ was compared to the RFU signal of the WT strain at the same dose. Statistical significance for this comparison is indicated as follows: ns ($P > 0.05$; non- significant), * ($P \leq 0.05$). **B. Δ$djlA$ responds to A22 and Mecillinam:** For the P$_{rprA}$::mCherry assay, the strains WT (EAW8) and $djlA$::$kan$ (AP46) were grown in MOPS minimal glucose medium and treated with nothing (uninduced) or with PMBN (20 µg/ml), MreB inhibitor A22 (5 µg/ml), or Mecillinam (0.3 µg/ml, Mec) at 37˚C. The RFU at OD 0.4 as

compared to WT uninduced, set to 1, is plotted. For statistical analysis, the RFU with each chemical treatment in Δ*djlA* was compared to the RFU signal of the WT strain for that compound. **C. PMBN response in Δ*djlA* at 42˚C:** For the P*rprA*::mCherry assay, the strains WT (EAW8) and *djlA*::*kan* (AP46) were grown in MOPS minimal glucose medium with or without PMBN (20 μg/ml) at 42˚C. The RFU at OD 0.4 as compared to WT uninduced is plotted. The statistical analysis shows the comparison between signals of Δ*djlA* and WT. **D. IgaA-RcsD interactions in Δ*djlA*:** The IgaA-RcsD interaction was tested using BACTH in BTH101 (WT) and AP 63 (*BTH 101 ΔdjlA*). Plasmids used were pEAW1 (IgaA-T18), pEAW8 (RcsD-T25), pEAW2 (IgaA-T25), pEAW7 (RcsD-T18), pEAW8peri (RcsD Δperi-T25), and pEAW8T (T411A). IgaA-RcsD interaction in WT was normalized to 1 and the interaction in Δ*djlA* (AP63) is plotted relative to this interaction. The interaction of IgaA-RcsD was 578 units, while the vector control was 25; these units are 1000x the slope of OD420 (see Materials and Methods). Statistical analysis using multiple unpaired *t*-tests is indicated as follows: ns ($P > 0.05$; non- significant), * ($P \leq 0.05$).
(TIF)

**S7 Fig. Characterization of DrpB mutants. A. Signaling by DrpB mutants in Δ*rcsF*:** For the P*rprA*::mCherry assay, the *rcsF*::*kan* (AP51) strain was transformed with the pBAD33 vector or pBAD33 plasmids carrying DrpB (pDSW1977), DrpB Δperi$_{48-57}$ (pAP3301), DrpB C29A (pAP3304), DrpB R38F (pAP3305), DrpB G83A (pAP3306), or DrpB T89A (pAP3307). The transformed strains were grown in MOPS minimal glycerol medium containing chloramphenicol (25 μg/ml) and either 0.2% glucose or 0.02% arabinose at 37˚C. The RFU at OD 0.4 is plotted relative to the uninduced WT, set to 1. **B. Analysis of DrpB mutants as *ftsEX* suppressors:** The strains WT (EC251) and Δ*ftsEX* (EC1215) were transformed with the same plasmids as in A and grown overnight in LB Miller media with chloramphenicol at 37˚C. The cultures were then normalized to OD$_{600}$ 1 and 4 ul dilutions were spotted on LB Miller (permissive), LBON (LB without NaCl) or LBON (with 0.2% arabinose). Plates were imaged after 16h incubation at 37˚C.
(TIF)

**S8 Fig. Analysis of DrpB-dependent signaling. A. DrpB interactions with Rcs components:** The interaction of DrpB with IgaA, RcsD, and RcsC was tested using the BACTH assay in BTH101. The IgaA-RcsD interaction was normalized to 1 and the other interactions were plotted relative to this interaction. The interaction of IgaA-RcsD was 490 units, while the vector control was 23; these units are 1000x the slope of OD420 (see Materials and Methods). No significant interaction of RcsC was observed with RcsD or DrpB. Plasmids used were pEAW2 (IgaA-T25), pEAW8 (RcsD-T25), pEAW6 (RcsC-T25), pEAW7 (RcsD-T18), pAP407 (DrpB-T18), pAP408 (DrpB-T25), and pKNT25 vector. Values (Mean ± SD) from independent experiments were statistically analyzed using multiple unpaired *t*-tests. 'ns' indicates a *P*-value > 0.05 (non- significant). **B. Rcs signaling by Δ*drpB*:** For the P*rprA*::mCherry assay, the strains WT (EAW8) and *drpB*::*kan* (AP41) were grown in MOPS minimal glucose medium and treated with nothing (uninduced) or with PMBN (20 μg/ml), MreB inhibitor A22 (5 μg/ml), or Mecillinam (0.3 μg/ml, Mec) at 18˚C. The RFU at OD 0.4 as compared to WT uninduced, set to 1, is plotted. The statistical analysis compared the signals of Δ*drpB* and WT for each condition. **C. DrpB signaling is independent of the *psp* pathway.** For this P*rprA*::mCherry assay, the WT (EAW8), *pspC*::*kan* (AP113) and *pspF*::*kan* (AP114) strains transformed with pBAD33 vector or pDrpB (pDSW1977) were grown in MOPS minimal glycerol medium containing chloramphenicol (25 μg/ml) either 0.2% glucose or 0.02% arabinose at 37˚C. The RFU at OD 0.4 is plotted relative to uninduced WT, set to 1. Statistical analysis was done using multiple unpaired *t*-tests and *P*-value for comparison with the uninduced vector

control of each strain is indicated as follows: ns ($P > 0.05$; non- significant) and ***
($P \leq 0.001$).
(TIF)

**S9 Fig. Signaling by multicopy YmgB.** For the $P_{rprA}$::mCherry assay, strains overexpressing
the pBAD33 vector (V) or pBAD33-YmgB (pAP3325) were grown in MOPS minimal glycerol
medium containing chloramphenicol (25 μg/ml) and either 0.2% glucose or 0.02% arabinose
at 37˚C. The RFU at OD 0.4 relative to the uninduced vector control, set to 1, is plotted here.
The strains used are: WT (EAW8), *rcsF*::*kan* (AP51), *rcsCΔperi* (EAW70), and *rcsD T411A*
(EAW121).
(TIF)

**S10 Fig. Signaling in *barA* deletion strains. A. Effect of *barA* deletion on Rcs signaling:** For
the $P_{rprA}$::mCherry assay, the cells were grown in MOPS minimal glucose medium at 37˚C.
The RFU at OD 0.4 as compared to WT, set to 1, is depicted here. The strains used were: WT
(EAW8), *barA*::*kan* (AP200), Δ*rcsC* (EAW91), Δ*rcsC barA*::*kan* (AP201), Δ*rcsD* (EAW19),
Δ*rcsD barA*::*kan* (AP202), *rcsB*::*Tn10* (AP50), and *rcsB*::*Tn10 barA*::*kan* (AP205). **B. DjlA and
DrpB can activate Rcs in Δ*barA*:** For the $P_{rprA}$::mCherry assay, the WT strain (EAW8) and
the *barA*::*kan* strain (AP200) containing either the vector or plasmids expressing DjlA (pDjlA/
pPSG961) or DrpB (pDrpB/pDSW1977) under the control of the arabinose-inducible pBAD
promoter were grown in MOPS minimal glycerol medium containing chloramphenicol
(25 μg/ml) and either 0.2% glucose or 0.02% arabinose at 37˚C. The RFU at OD 0.4 compared
to the WT vector uninduced control, set to 1, is plotted. **C. DsbA signaling in Δ*barA*:** For this
$P_{rprA}$::mCherry assay, the strains were grown in MOPS minimal glucose medium at 37˚C. The
RFU at OD 0.4 as compared to WT uninduced, set to one, is shown here. The cells were sub-
jected to either 1mM DTT or 20μg/ml PMBN. The strains used were: WT (EAW8), *barA*::*kan*
(AP200), Δ*dsbA* (AP11), and Δ*dsbA barA*::*kan* (AP206). Statistical significance was calculated
using multiple unpaired *t*-tests and is shown as follows: ns ($P > 0.05$; non- significant), *
($P < 0.05$), ** ($P \leq 0.01$), and *** ($P \leq 0.001$).
(TIF)

## Acknowledgments

We thank members of the Gottesman lab for comments on the manuscript. Tae Yoon Chang,
a summer student, performed the initial library screen for RcsF-independent Rcs activation
that identified DrpB as an activator. We thank David Weiss for discussions and for providing
strains (see S1 Table).

## Author Contributions

**Conceptualization:** Anushya Petchiappan, Nadim Majdalani, Erin Wall, Susan Gottesman.

**Formal analysis:** Anushya Petchiappan.

**Funding acquisition:** Susan Gottesman.

**Investigation:** Anushya Petchiappan, Nadim Majdalani, Erin Wall.

**Project administration:** Susan Gottesman.

**Resources:** Nadim Majdalani.

**Supervision:** Susan Gottesman.

**Validation:** Anushya Petchiappan.

**Writing – original draft:** Anushya Petchiappan, Susan Gottesman.

**Writing – review & editing:** Anushya Petchiappan, Nadim Majdalani, Susan Gottesman.

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
