## [Decision Letter · Decision Letter 0]

27 Sep 2024

Dear Dr Gottesman,

Thank you very much for submitting your Research Article entitled 'RcsF-independent mechanisms of signaling within the Rcs Phosphorelay' to PLOS Genetics.

The manuscript was fully evaluated at the editorial level and by independent peer reviewers. The reviewers appreciated the attention to an important problem, but raised some substantial concerns about the current manuscript. Based on the reviews, we will not be able to accept this version of the manuscript, but we would be willing to review a revised version. We cannot, of course, promise publication at that time.

If you decide to revise the manuscript for further consideration at PLOS Genetics, please aim to resubmit within the next 60 days, unless it will take extra time to address the concerns of the reviewers, in which case we would appreciate an expected resubmission date by email to plosgenetics@plos.org.

To resubmit, log into your Editorial Manager account and select the option 'Revise Submission' in the 'Submissions Needing Revision' folder.

We are sorry that we cannot be more positive about your manuscript at this stage. Please do not hesitate to contact us if you have any concerns or questions.

Yours sincerely,

Aretha Fiebig, PhD

Academic Editor

PLOS Genetics

Lotte Søgaard-Andersen

Section Editor

PLOS Genetics

The reviewers were in generally enthusiastic about the work presented, and in agreement that it represents a significant advancement in our understanding of the Rcs system in E. coli. Reviewers also noted that the manuscript was generally well written. However, there were some concerns, most of which can be addressed with revisions to the text. 

Reviewer's Responses to Questions

**Comments to the Authors:**

Reviewer #1: In this manuscript, Petchiappan and colleagues unveil three distinct pathways of Rcs activation that are independent of the sensor protein RcsF, that is the mutation of dsbA, the overexpression of the DnaK cochaperone DjlA, and of the inner membrane protein DrpB, using a combination of genetics and phenotypic approaches. The findings of the study undoubtedly advance the global understanding of Rcs signaling, which is rather complex, and provide additional insights on the implication of the different Rcs components and of their interaction in integrating different input signals.

The experiments are properly designed with the proper controls and the conclusions well supported with compelling data. Although the manuscript is quite dense because of the large amount of data, it is very well written and results are well interpreted and analyzed, thereby supporting the proposed model.

I have few comments, mostly minor, and some questions that I recommend the authors to address to improve the manuscript.

Comments

1)Introduction.

An RcsF-independent pathway of Rcs activation was identified in Salmonella (Salvail and Groisman, 2020; PMID: 32392214) in which BarA promotes RcsB activation during exponential growth in LB. The authors should mention these findings in the introduction.

2) Results. Lines 166-167. “[…] while deletion of rcsC caused an increase in the basal level of PrprA::mCherry expression […]”

Is the fusion activity increased because of the absence of RcsC’s phosphatase activity? The authors should provide a short interpretation of the phenotype in the text.

3) Figure S4E.

Why is dsbA mutation not resulting in an increase of fusion activity in the absence of inducer (uninduced IgaA dsbA+ compared with uninduced IgaA delta dsbA)? The strain is delta rcsD but was complemented with a plasmid expressing rcsD, which should recover Rcs signaling.

4) Results. Line 368.

What was the goal of doing the assay at high temperature? The authors should briefly explain the rationale of the experiment.

5) Figure S6A.

Is DjlA expressed under the tested conditions? If possible, authors should provide evidence from their own data or from the literature that the protein is produced under the growth conditions tested. The absence of phenotype for the djlA mutation could be due to DjlA not being expressed, or not sufficiently, under the conditions in which the experiment was performed. That the djlA mutation has an effect on IgaA-RcsD interaction in Fig. S6C suggests that DjlA is produced, but one must consider that the strain and growth conditions used for the BACTH assay are different than the ones used in Fig. S6A.

6) Figure S8B.

Is DrpB expressed under the growth conditions of the assay? If possible, authors should provide evidence from their own data or from the literature that the protein is produced under the growth conditions tested. The absence of phenotype for the drpB mutation could be due to DrpB not being expressed, or not sufficiently, under the conditions in which the experiment was performed.

7) Results

As BarA was previously established as a RcsF-independent activator of RcsB in Salmonella (Salvail and Groisman, 2020; PMID: 32392214), authors should determine whether the Rcs activation phenotypes observed upon DjlA or DrpB overexpression or dsbA inactivation also occur in a barA mutant to rule out whether those are BarA-dependent or not, assuming that BarA also activates RcsB in E. coli.

Minor Comments

1) Author summary. Line 41.

Replace “Enterobacteria” with “enterobacteria” to be consistent with the way it is written in the rest of the manuscript.

2) Results. Line 129.

A reference should be added to support the information provided on Spot42.

3) Results. Lines 144-145.

The statement should be supported by a reference.

4) Results. Line 247.

Replace “Bordetella” with “Bordetella pertussis”.

5) Results. Line 248.

Add “E. coli” before “strain”.

6) Results. Line 281.

Add “data not shown” after “mucoid”.

7) Results. Lines 387-388. “DrpB localizes to the divisome and overexpression allowed the �ftsEX.”

This sentence seems incomplete and should be revised.

8) Results. Line 438.

Replace “targts” with “target”.

9) Discussion. Line 545.

Add a space between “by” and “DrpB”.

Reviewer #2: The Rcs stress response is a very complex (the most complex identified so far?) envelope stress response in Gram-negative bacteria. The RcsBCDF-IgaA factors were identified many years ago. Even though significant advances in understanding this system have been made, we still know very little about how signals are sensed and transmitted in the Rcs system. This work by Petchiappan et al. addresses this problem. Using solid, thoroughly applied, classical (yes!) genetic approaches, the authors found that overproduction of the division protein DrpB activates Rcs. They further characterized requirements in the Rcs system for this activation. The authors also characterized other known “signals”, mainly the overproduction of the chaperone DjlA and the loss of the DsbA periplasmic oxidoreductase. In addition to providing mechanistic insights into how these signals are sensed and transmitted, the work is first to show experimental evidence supporting that the RcsC histidine kinase acts as a sensor for at least some signals. Previously, the only bonified sensor in Rcs was the outer membrane lipoprotein RcsF.

This study addresses a fundamental question (how signals are sensed and transmitted) relevant to the bacterial cell envelope, signal transduction, and physiology. Answering this question is often the major limitation in understanding stress responses. The work was very well designed and executed. The authors were very thorough. The manuscript was very well crafted, and the conclusions presented by the authors are justified. It was a pleasure to read. Well done! As a reviewer, I truly appreciate the high quality of the data and manuscript. I only have a minor point that I think the authors should address, as well as two corrections.

Minor point:

1) The loss of activation by the DjlA-TM variant could be the result of a loss of protein localization and/or protein stability (maybe the protein is not stable if not anchored to the membrane). This should be clearly stated when discussing the relevant results in the Results and Discussion sections unless the authors can rule out issues with protein levels through immunoblotting (not required to include these data, text clarification would suffice).

Text corrections:

Line 252: "We introduced the dsbA null mutant into the host for the BACTH assay" should be "We introduced the dsbA null mutant allele into the host for the BACTH assay".

Line 406: "DrpB mutants" should be "drpB (italics) mutants”

Reviewer #3: Regulator of Capsule Synthesis (Rcs) is a complex stress response system involving the RcsCDB phosphorelay and additional regulators, including RcsF and IgaA. Many different conditions induce Rcs, but the signal and the molecular mechanism of activation remain largely unknown. While most conditions require the sensory component RcsF to alleviate phosphorelay inhibition by IgaA, there are known genetic conditions that can lead to activated Rcs even in the absence of RcsF, such as ΔdsbA and DjlA overexpression. In this manuscript, the authors report an additional condition (DrpB overexpression) identified in a screen and characterize all three condition for potential mechanisms of action. They show that ΔdsbA and DjlA overexpression likely interfere with IgaA/RcsD interaction by modulating protein folding, while DrpB overexpression induces Rcs via a different mechanism, likely by generating a signal recognized by the periplasmic domain of the RcsC kinase. Overall, this report adds yet another level of complexity to Rcs and provides evidence for at least some sensory function of the RcsC kinases.

Major comments:

My first comment is related to semantics. The authors refer several times in the manuscript to these conditions as distinct “signaling pathways.” Personally, I find this confusing, as the authors show that activation of Rcs is a result of the canonical RcsCDB phosphorelay, so the signal transduction/phosphorylation pathway did not change. It may also be misleading to call them “signals” as it may imply physiological signals for the Rcs system. For example, the authors argue that ΔdsbA results in a misfolded IgaA due to the reduction of disulfides rather than generating a “signal”. Essentially, Rcs activation is a result of malfunction of a Rcs component. As the authors discuss, it is not yet clear whether these conditions are physiologically relevant. So, I think the text should be carefully revised to find a better way to describe the reasons behind changes in Rcs activity.

I don’t agree with the conclusion in lines 174-175 that the effect of ΔdsbA is independent of the RcsC periplasmic domain, as deletion of rcsCΔperi in the ΔdsbA background leads to a dramatic reduction of Rcs activity. It is possible that because ΔdsbA is quite pleiotropic, it acts both directly on IgaA but also creates an unknown “signal’ recognized by the RcsC periplasmic domain. Alternatively, there may be some effects of rcsCΔperi on IgaA/RcsD regulation. After all, the authors argue that Rcs signaling can be induced through the RcsC periplasmic domain, and hence must be able to overcome the IgaA/RcsD checkpoint. To this end, is there any effect of rcsCΔperi and ΔdsbA mutations in the igaA C4S background? This is a simple experiment that may help clarify the role of the RcsC periplasmic domain.

Additional comments:

Line 222: Remove RcsD.

Lines 360-361: How can DjlA regulate cytoplasmic interaction between IgaA/RcsD if it has an effect on IgaAΔcyto/RcsD strain, in which there is no cytoplasmic interaction?

An alternative explanation for DjlA is that its overexpression may interfere with membrane insertion, for example, of IgaA. This would explain why RcsD T411A does not have an effect in this background. I think there were some reports about DnaK modulated membrane protein transport.

Line 490: Oxidative stress typically promotes cysteine oxidation. If anything, it is under anaerobic conditions combined with a decrease in DsbA activity that can lead to disulfide reduction.

I think the manuscript may benefit from some polishing of the writing and the flow. There are some parts that are repetitive. My suggestion is to move the DrpB screen section toward the end, ahead of the DrpB mechanism section, as these sections appear to be disconnected in the current version.

There are important statements in the introduction and throughout the paper that don’t have any references.

**Have all data underlying the figures and results presented in the manuscript been provided?**

Reviewer #1: Yes

Reviewer #2: Yes

Reviewer #3: Yes

PLOS authors have the option to publish the peer review history of their article (what does this mean?). If published, this will include your full peer review and any attached files.

Reviewer #1: No

Reviewer #2: No

Reviewer #3: No

---

## [Decision Letter · Decision Letter 1]

5 Dec 2024

Dear Dr Gottesman,

We are pleased to inform you that your manuscript entitled "RcsF-independent mechanisms of signaling within the Rcs Phosphorelay" has been editorially accepted for publication in PLOS Genetics. Congratulations!

Yours sincerely,

Aretha Fiebig, PhD

Academic Editor

PLOS Genetics

Lotte Søgaard-Andersen

Section Editor

PLOS Genetics

Aimée Dudley

Editor-in-Chief

PLOS Genetics

Anne Goriely

Editor-in-Chief

PLOS Genetics

Comments from the reviewers (if applicable):

Thank you for your thorough consideration of the referees' comments. The reviewers were all satisfied with the revisions and all agreed the manuscript is improved and suitable for publication. Congratulations!

Reviewer's Responses to Questions

**Comments to the Authors:**

Reviewer #1: The authors thoroughly addressed all my comments. I am satisfied with this improved version of the manuscript.

Reviewer #2: The authors have modified their manuscript to adequately address previous review. Overall, the manuscript has been improved.

This work advances our understanding of the complex Rcs system. The authors' conclusions are justified and the overall quality of the manuscript is high. Well done!

Reviewer #3: My comments were appropriately addressed, and I am satisfied with revision.

**Have all data underlying the figures and results presented in the manuscript been provided?**

Reviewer #1: Yes

Reviewer #2: Yes

Reviewer #3: Yes

PLOS authors have the option to publish the peer review history of their article (what does this mean?). If published, this will include your full peer review and any attached files.

Reviewer #1: No

Reviewer #2: No

Reviewer #3: No

**Data Deposition**

http://datadryad.org/submit?journalID=pgenetics&manu=PGENETICS-D-24-00984R1

**Press Queries**

---

## [Editor Report · Acceptance letter]

17 Dec 2024

PGENETICS-D-24-00984R1 

RcsF-independent mechanisms of signaling within the Rcs Phosphorelay 

Dear Dr Gottesman, 

We are pleased to inform you that your manuscript entitled "RcsF-independent mechanisms of signaling within the Rcs Phosphorelay" has been formally accepted for publication in PLOS Genetics! Your manuscript is now with our production department and you will be notified of the publication date in due course.

With kind regards,

Zsofia Freund

PLOS Genetics

On behalf of:
